# $(\text{Implicit})^2$ : Implicit Layers for Implicit Representations

**Zhichun Huang**
Carnegie Mellon University
Pittsburgh, PA 15213
zhichunh@cs.cmu.edu

**Shaojie Bai**
Carnegie Mellon University
Pittsburgh, PA 15213
shaojieb@cs.cmu.edu

**J. Zico Kolter**
Carnegie Mellon University
Pittsburgh, PA 15213
zkolter@cs.cmu.edu

## Abstract

Recent research in deep learning has investigated two very different forms of "implicitness": *implicit representations* model high-frequency data such as images or 3D shapes directly via a low-dimensional neural network (often using e.g., sinusoidal bases or nonlinearities); *implicit layers*, in contrast, refer to techniques where the forward pass of a network is computed via non-linear dynamical systems, such as fixed-point or differential equation solutions, with the backward pass computed via the implicit function theorem. In this work, we demonstrate that these two seemingly orthogonal concepts are remarkably well-suited for each other. In particular, we show that by exploiting fixed-point *implicit layer* to model *implicit representations*, we can substantially improve upon the performance of the conventional explicit-layer-based approach. Additionally, as implicit representation networks are typically trained in large-batch settings, we propose to leverage the property of implicit layers to *amortize* the cost of fixed-point forward/backward passes over training steps – thereby addressing one of the primary challenges with implicit layers (that many iterations are required for the black-box fixed-point solvers). We empirically evaluated our method on learning multiple implicit representations for images, audios, videos, and 3D models, showing that our $(\text{Implicit})^2$ approach substantially improve upon existing models while being both faster to train and much more memory efficient.

## 1 Introduction

The concept of "implicitness" has recently been applied to various contexts of machine learning research. One particular thread is *implicit representations* [19, 22, 28, 27], which aims to learn a continuous representation of high-frequency, often discretely-measured signals such as large images. Formally, given a spatial-temporal coordinate input $x \in \mathbb{R}^d$, an implicit representation of it is a function $\Phi : \mathbb{R}^d \to \mathcal{C}$, where $\mathcal{C}$ is the space of desired quantity (e.g. color, volume density, distance, etc.) and $\Phi$ is usually parameterized with a neural network. Such implicitness has multiple benefits over the conventional discrete (e.g., grid-based) representations; e.g., as $\Phi$ is defined on a continuous domain, an implicitly represented input consumes much less storage than the original input; as another example, $\Phi$ is differentiable, hence allowing for the computation of higher-order derivatives. However, the training of these $\Phi$ networks themselves are usually quite memory-consuming, since we usually deal with very high-resolution images and videos, and training is typically done in full-batch mode (e.g., when training on a batch size of $512 \times 512$, a simple 4-layer MLP with 1024 hidden units already requires $> 16$ GB memory just to store the intermediate activations).

A recent (orthogonal) usage of *implicitness* in deep learning comes from the notion of *implicit layers/models*, where the word is used to characterize model architectures (rather than input representations). The nomenclature comes from the concept of implicit vs. explicit functions: instead of

35th Conference on Neural Information Processing Systems (NeurIPS 2021).

representing a model as an explicit stacking of layers, an implicit model solves a non-linear dynamical system [10] (e.g., ODEs [7, 9] or fixed-points [1, 32]). Importantly, these implicit models decouple the forward and backward passes and analytically differentiate via the implicit function theorem [14]. This enables these models to use *constant training memory*, irrespective of the forward pass trajectory. Moreover, recent works on deep equilibrium models [1] have been shown to be able to achieve results on par with the state-of-the-art explicit models (e.g., Transformers [31]) on very large-scale vision or NLP tasks [1, 2]. However, despite training memory efficiency, these implicit layers are also slower to compute (e.g., by 2-3$\times$), since the solvers are usually iterative in nature and ultimately require many function evaluations.

Despite the success of these "implicit" methods in their respective areas, the two concepts so far have rarely intersected: existing models used for implicit representations are neither *implicit* in nature (since they are still stacked multi-layer explicit operators trained end-to-end in the feedforward manner), nor do they exploit any properties of the implicit functions during training (e.g., implicit differentiation). In this paper, we argue that implicit representations and implicit layers are remarkably well-suited to each other, and their use together can compensate for their aforementioned drawbacks.

Specifically, we propose to combine the best of both worlds by replacing the explicit models used in conventional *implicit representation* learning with implicit layers, whose output is defined by a *fixed-point condition*, thus dubbed (Implicit)$^2$ networks. Formally, let $x$ be the input and $F$ be an (often shallow) layer, the (Implicit)$^2$ approach learns the following implicit representation $\Phi$:

$$\Phi(x) = Wz^\star + b, \text{ where } z^\star = F(z^\star; x) \tag{1}$$

Importantly, we show that these two "implicitness" complement each other well, especially in two important aspects. First, the unique large-batch training scheme of *implicit representations* as well as the forward/backward decoupling properties of *implicit models* permit us to *amortize* the cost of the iterative solver that would otherwise make implicit models slow to train. We show that by fixed-point reuse (in the forward pass) and Jacobian approximation (in the backward pass), training an implicit model on implicit representation tasks requires almost no computation overhead. Second, implicit models allow us to train these high-frequency input tasks with a much lower memory consumption while not losing speed or performance, thus significantly lowering the hardware requirement and computation budget of training implicit representations. We evaluate (Implicit)$^2$ on multiple implicit representations for images, videos, and audios, showing substantial improvement upon existing competitive explicit-model methods like SIREN [27] or MFN [11] using equivalent-sized networks.

## 2 Background and Preliminaries

### 2.1 Implicit Neural Representations (INR)

While high-frequency data such as images and scene geometry have been traditionally represented discretely (e.g., pixel/voxel grids or mesh points), recent work has demonstrated the possibility of replacing them with continuous functions parameterized by multi-layer perceptrons (MLPs) [27, 28]. These deep networks have been used to learn differentiable representations of various forms, such as continuous spatial-temporal image representations [5, 8], high-resolution scene representations [26, 22], and volumetric rendering [19, 29], and have been shown to be significantly more compact than the conventional grid-based approaches.

However, training these INR models is not easy. For example, the training of implicit representations is usually conducted in very large batch sizes (e.g., $512 \times 512$), which renders these (albeit simple) MLPs very memory-consuming during training. As another example, recent works have shown that typical deep networks with ReLU/tanh/sigmoid non-linearities fail to capture the fine details of an input signal, and resorted to periodic non-linearities [27], Fourier feature encodings [28], or repeated application of non-linear filters (e.g., Gabor wavelets) to the input and then multiply with the features [11]. We introduce below the two latest, highly competitive models that have been developed on this task, which our work will build on (see §3.1).

**Sine-activated Network for Implicit Representations (SIREN).**    To overcome the detrimental effect of traditional non-linearities like ReLU/tanh on modeling fine details and higher-order derivative of the input signals, Sitzmann et al. [27] proposes to use sinusoidal activation functions that allow explicit supervision on any derivatives of the input signal. The resulting MLPs, though simple, have been shown to achieve state-of-the-art performance in representing images, videos, complex

geometries, and more [27]. Formally, given an input $x$, SIREN learns an implicit representation by the following $L$-layer MLP:

$$\Phi(x) = W(g_{L-1} \circ g_{L-2} \circ \cdots \circ g_0)(x) + b \tag{2}$$

where $g_i$ is the $i$-th layer of the network that computes $g_i(h^{[i]}) = \sin(W_i h^{[i]} + b_i)$ with a sine non-linearity, and $h^{[i]}$ denotes the hidden state at $i$-th layer.

**Multiplicative Filter Networks (MFN).** More recently, Fathony et al. [11] proposed to forego the SIREN-like design (i.e., MLPs with periodic activation functions) in favor of multiplicative operations that are proven to be similarly expressive. Specifically, at each layer $i$ of the network, an MFN applies a learnable non-linear filter kernel (parameterized by $\theta^{(i)}$) $g(x; \theta^{(i)})$ on the *original input* $x \in \mathbb{R}^n$, and elementwise multiply it with a *linear transformation* of the features; i.e., formally,

$$\Phi(x) = W h^{[L]} + b, \quad \text{where } h^{[i+1]} = \left( W_i h^{[i]} + b_i \right) \cdot g(x; \theta^{(i)}) \text{ and } h^{[0]} = g(x; \theta^{(0)}) \tag{3}$$

Fathony et al. [11] in particular proposed two instantiations that admit sinusoids or a Gabor wavelet as the filter $g$, which are called FourierNet and GaborNet, respectively:

$$
\begin{aligned}
g_{\text{Fourier}}(x; \theta^{(i)}_{\text{Fourier}}) &= \sin(\omega^{(i)} x + \phi^{(i)}) \\
g_{\text{Gabor}}(x; \theta^{(i)}_{\text{Gabor}}) &= \sin(\omega^{(i)} x + \phi^{(i)}) \cdot \exp\left( -(\gamma^{(i)}/2) \cdot (x - \mu^{(i)})^2 \right)
\end{aligned}
\tag{4}
$$

where $\theta^{(i)}_{\text{Gabor}} = \{\omega^{(i)} \in \mathbb{R}^n, \phi^{(i)} \in \mathbb{R}, \gamma^{(i)} \in \mathbb{R}, \mu^i \in \mathbb{R}^n\}$ (similar for $\theta^{(i)}_{\text{Fourier}}$), and $(\mu^{(i)}, \gamma^{(i)})$ denotes the mean and scale of the Gabor filter. These models have shown performance on par with or better than the periodic-nonlinearity-based networks like SIREN on a range of tasks like image and video representations.

## 2.2 Implicit Models

Whereas explicit models hierarchically stack a sequence of $L$ non-linear operators to build a computation graph for their forward/backward passes (where $L$ is often referred to as the *depth* of a model), implicit model instead define the output as a solution to a non-linear dynamical system which in most cases can't be expressed as a fixed computation graph. For example, the Neural ODE model [7] solves an initial value problem ODE that intuitively corresponds to infinitesimal residual layers [13]. Another example is the recent deep equilibrium (DEQ) models [1], whose forward pass involves computing the fixed point of a shallow layer $F$.

Importantly, unlike explicit models, these implicit models do not need to store any intermediate activations during training: to produce gradients in the backward pass, we can directly differentiate through their *final outputs* (e.g., by adjoint method for Neural ODEs [7] and implicit differentiation for equilibrium models [1]), regardless of how the forward pass was computed. Therefore, the forward and backward passes of implicit models can rely on independent black-box solvers, and these models consume only a constant training memory. Our work is primarily built on the formulation of deep equilibrium models, which we introduce in detail below.

**Deep Equilibrium (DEQ) Models.** Given an input $x$ and a layer $F(z; x)$ (e.g., a residual block, or a multi-head self-attention layer [31]), a DEQ model aims to approximate an infinite-level stacking of this layer (i.e. $z^{[i+1]} = F(z^{[i]}, x)$ with $i = 1, \ldots, L$ and $L \to \infty$). To achieve this purpose, Bai et al. [1] advocated for directly solving a fixed-point condition defined by $F$ and $x$:

$$z^\star = F(z^\star; x) \tag{5}$$

where $z^\star$ is called the equilibrium since, intuitively, stacking another $F(\cdot, x)$ layer on top does not change the activation. To solve for the equilibrium point, prior works have had success with various kinds of advanced solvers, such as Broyden's method [6, 1, 17], and Peaceman-Rachford splitting [23, 32, 24]. Then, in the backward pass, we can produce the gradient by the implicit function theorem [14] directly applied on this equilibrium point $z^\star$, with a loss $\ell$:

$$\nabla_{(\cdot)} \ell = \frac{\partial \ell}{\partial z^\star} \left( I - J_F \right)^{-1} \frac{\partial F(z^\star; x)}{\partial (\cdot)} \tag{6}$$

where $J_F$ denotes the Jacobian of $F$ evaluated at $z^\star$. As the inverse of a matrix is expensive to compute, in practice the backward pass is usually formulated as another *linear* fixed-point condition

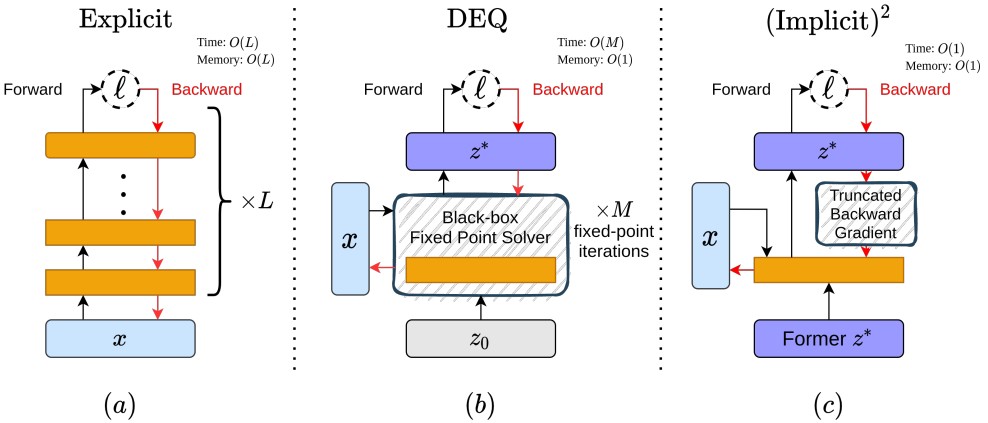

Figure 1: (a) Explicit networks incur $O(L)$ complexity in both time and memory during training. (b) Original DEQ models requires constant memory, yet they take $O(M)$ time in forward/backward passes, where $M$ is the number of fixed-point solver steps. (c) In contrast, an (Implicit)$^2$ Network consumes constant amortized memory *and* time in both forward and backward passes.

in $y$: $y^\top J_F + \frac{\partial \ell}{\partial z^\star} = y^\top$, which again can be solved via any of the advanced black-box solvers. Thus theoretically, the model only needs to store $x$, $z^\star$ and $F(z^\star; x)$ in memory at training time, which is equivalent to the memory consumption of merely one $F$ layer. Recent works [1, 2] have demonstrated that DEQ models can scale to high-dimensional tasks and be as competitive as explicit structures like ResNet [13] and Transformers [31] on large-scale computer vision and NLP tasks. However, these implicit models are also slower (typically by $\sim 3\times$) to train, because both forward and backward passes involve numerous iterative steps.

Our work studies the novel combinations of *implicit models* and *implicit representation learning*. Specifically, we show that these two "implicitness" are surprisingly well-suited to each other, as they almost perfectly compensate for each others' drawbacks. We demonstrate that implicit modeling of a simple layer $F$ substantially improves the training speed, memory, *and* performance on implicit representation tasks when compared to the aforementioned state-of-the-art deep explicit networks.

## 3  (Implicit)$^2$ Networks

As we discussed in §2.1, learning implicit representation for complex signals is often more effective when projecting the input coordinates onto high-frequency bases (e.g. sinusoidal functions), making the designs of these models typically different from deep networks used in more common applications, like image classifications (e.g., ResNets [13]). In this work, we propose to directly build on the designs of these aforementioned state-of-the-art layer architectures by only making minimal modifications, but modeling them as shallow implicit models. We briefly introduce below the two instantiations of our proposed (Implicit)$^2$ approach, which are based on SIREN [27] and MFN [11], respectively, followed by a discussion of how implicit representation tasks can enable us to amortize the cost of training these implicit networks significantly in practice.

### 3.1  (Implicit)$^2$ Network Architctures

**Implicit Sine-activated Networks (iSIREN).**    Motivated by the success of SIREN, we first propose an *input-injected* variant of SIREN (denoted as *SIREN (input inj.)*) suitable for implicit models, i.e.,

$$F_{\text{SIREN}}(z; x) = \sin(W(z + \sin(Vx)) + Ux + b) \tag{7}$$

where, following prior works on deep equilibrium models [1, 32, 24], the transformed input $x$ is added to the hidden representation $z$ (see Fig. 2). Just like the canonical SIREN layer, any order of derivative of this input-injected SIREN variant is also an input-injected SIREN of the same form. Therefore, the proposed variant inherits the same high-frequency and higher-order differentiability properties as SIREN, and we name the implicit model that solves its fixed-point as *iSIREN*.

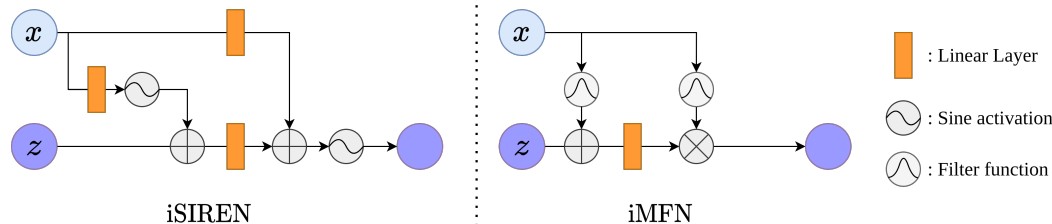

iSIREN       iMFN

           : Linear Layer

           : Sine activation

           : Filter function

Figure 2: The architecture of one implicit layer $F(z; x)$ in iMFN and iSIREN. Note that when $z = 0$, evaluating $F(z; x)$ once is equivalent to the output of the respective explicit layer.

**Implicit Multiplicative Filter Networks (iMFN).** Additionally, we introduce a variant based on the recent Multiplicative Filter Networks (MFN) [11]. As MFNs only perform linear transformations on the hidden features, which are multiplied with a non-linear filter function on the original input $x$, we slightly modify the layer design as follows to introduce an additional input injection:

$$F_{\text{MFN}}(z; x) = (W(z + g(x; \theta_2)) + b) \circ g(x; \theta_1) \tag{8}$$

where $g$ denotes the filter function of choice, such as Gabor or Fourier ($g_{\text{Gabor}}$ and $g_{\text{Fourier}}$ in Eq. (4)) filters [11]. We call the implicit models defined by this layer iMFN. Moreover, just like the original MFN [11], we can show that the output of iMFN (which is the fixed-point of $F_{\text{MFN}}$) is also a linear combination of the non-linear filter functions (see Appendix **??** for proof).

We provide visualizations of these two architectural instantiations in Fig. 2. With these variants, we formally introduce the corresponding (Implicit)$^2$ approaches to implicit representation tasks as follows:

$$\Phi(x) = W'z^\star + b', \text{ where } z^\star = F(z^\star; x), \text{ and } F \in \{F_{\text{MFN}}, F_{\text{SIREN}}\} \tag{9}$$

## 3.2 Accelerated Training of (**Implicit**)$^2$ Networks

Compared to explicit networks, training with implicit models significantly lowers the *memory* budget required (thanks to implicit differentiation). This is especially compelling for training implicit representation tasks, which are often memory-bottlenecked. In this subsection, we describe how one can exploit 1) the large-batch nature of implicit representation training; and 2) the nature of fixed-point optimizations (in both forward and backward), to significantly improve the *speed* of these models as well, making them eventually both more memory-efficient *and* time-efficient than explicit models (while achieving performance as competitive, or better; see §4).

**Fixed-point Reuse.** Suppose we fix a training step $t$ and let the corresponding network parameter (i.e., weights of the network) be given by $\theta_{(t)}$. Typically, a deep equilibrium model solves for the fixed-point $z^\star_{(t)}$ given an input $x$ with a black-box fixed-point solver starting with an initial guess $z^{[0]}$ (which is usually $\mathbf{0}$ or a random vector in $\mathcal{N}(0, I)$):

$$z^\star_{(t)} = \textsf{FixedPointSolve}(F, \theta_{(t)}, x, z^{[0]}) \quad (10)$$

where we note that $z^\star_{(t)}$ does not depend at all on this initial guess $z^{[0]}$ of the optimization. Depending on the quality of the initial guess, the optimization process itself can take a different number of steps. For example, in the unlikely circumstance where we guessed $z^{[0]}$ to be $z^\star_{(t)}$, the fixed-point solvers will directly converge at

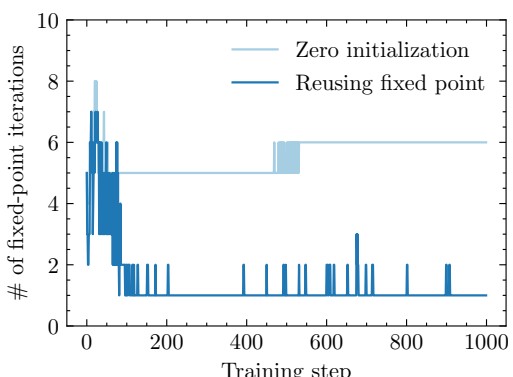

Figure 3: Zero initialization *v.s.* fixed-point reuse in terms of required steps to converge. Further details of this experiment can be found in §4.1.

iteration 0. However, in practice, this property is rarely used, as given a complex high-dimensional input $x$ (e.g., a high-resolution image), it is hard to know (or learn) what makes a good initial guess.

Therefore, these implicit models' aforementioned memory efficiency comes at a cost: without good initial guesses, we must spend more iterative updates in order to secure convergence to a good enough estimate of the fixed point in practice, rendering a slower model. However, we argue that the training on many implicit representation tasks, which uses the entire batch of data (e.g., to fit an RGB-color image implicit representation, each pixel of this image is a "sample" of this full batch), can compensate for this issue by providing us very reasonable initial guesses. In particular, training in the full-batch mode allows us to have direct access to *all* of the fixed-points $z^\star_{(t)}$ for all inputs $x$ in the training set immediately after step $t$. Assuming the model updates between iterations $t$ and $t+1$ are small (which is true in practice, such as via SGD with a small learning rate $\eta$), one can safely assume the fixed-points of the updated layer $F(\cdot; x, \theta_{(t+1)})$ do not deviate much from its current estimate $z^\star_{(t)}$. Formally, $F(\cdot; x, \theta_{(t+1)}) \to F(\cdot; x, \theta_{(t)})$ as learning rate $\eta \to 0$, which implies $z^\star_{(t+1)} \to z^\star_{(t)}$. We empirically verify this on these full-batch training settings (see Fig. 3), where a majority of the fixed-point convergences (to a target residual level $\varepsilon = 0.01$) finish in exactly 1 step.

However, we note that this does not imply all training iterations will gain a similar level of convergence boost (e.g., see the first 100 steps in Fig. 3). In particular, at training iteration $t = 0$, we do not have a "previous estimate" at all, which means we still have to solve for $z^\star$ by running the solver for several steps with a neutral initial guess. Thus, we highlight that such fixed-point reuse *amortizes* the cost of later training iterations, and indeed, the majority of the training of the implicit models. Empirically, we found that performing one step of the fixed-point iteration is sufficient for training the iSIREN and iMFN networks; i.e., at training step $t + 1$, we simply perform

$$z^\star_{(t+1)} = F(z^\star_{(t)}; x, \theta_{(t)}). \tag{11}$$

**Truncated Backward Gradient.** As mentioned in §2.2, the backward gradient of implicit models (Eq. (6)) can be alternatively obtained by solving a linear fixed-point system

$$y^\top = y^\top J_F + \frac{\partial \ell}{\partial z^*} \tag{12}$$

so as to avoid expliciting inverting the matrix. However, recent works have suggested that assuming sufficient stability in the DEQ model layer $F$, we can also approximate $\frac{\partial \ell}{\partial z^*} \left(I - J_F|_{z^*}\right)^{-1}$ by an expansion of it as a Neumann series [15]:

$$y^\top = \frac{\partial \ell}{\partial z^\star} \sum_{i=0}^{\infty} (J_F)^i \tag{13}$$

For example, Fung et al. [12] propose to use the 0-th order approximation and perform a Jacobian-free backward gradient estimation (assuming Lipschitz $F$); i.e., $y^\top \approx \frac{\partial \ell}{\partial z^\star}$. Although this approximated gradient was empirically shown to be effective in image classification tasks, we found it to substantially hurt the training-time error convergence in implicit representation learning. As a result, we propose to use a truncated backward gradient such that the Neumann series is unrolled for a $T > 0$ steps to better

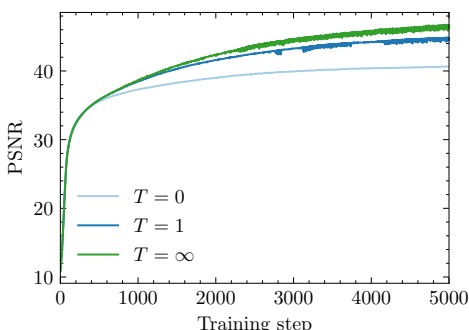

Figure 4: Training PSNR comparison between different levels of gradient approximation. $T$ denotes the approximation order as in Eq. 14, where $T = \infty$ indicates convergence to the solution.

approximate the exact implicit differentiation. Formally, the proposed gradient has the form

$$\hat{\nabla}_{(\cdot)} \ell = \frac{\partial \ell}{\partial z^*} \sum_{i=0}^{T} (J_F)^i \frac{\partial F(z^*; x)}{\partial (\cdot)}. \tag{14}$$

In practice, we found $T = 1$ to be already sufficient for implicit representation tasks (see Fig. 4), which requires only one evaluation of the vector-Jacobian product $\frac{\partial \ell}{\partial z^*} J_F$.

**Spectral Normalization.** We additionally apply spectral normalization on the layers $F$ we use for the implicit models, which ensures that $F$ (for forward passes) and $J_F$ (for backward passes) are contractive mappings, and is able to guarantee unique and stable fixed-points due to the Banach Fixed

Point Theorem [3]. Specifically, we adopt the power-iteration in [20, 4] to scale the spectral norms of $W$ in Eq. (7) and (8) after each training iteration if it become $> 1$, which incurs little additional computation cost [20]. However, we note that contractivity is not always necessary for fixed-point convergence (especially in the presence of stronger fixed-point solvers, as shown in large-scale DEQ models [1]), and weaker regularizers such as weight normalization [25] may also suffice in practice.

## 4 Experiments

We evaluate our (Implicit)$^2$ networks on several representation tasks and compare the difference in performance between the implicit and explicit modeling of implicit representations across a variety of configurations. For both explicit networks and implicit networks, we use $L$ to denote the number of layers and $D$ to denote the hidden dimensionality of the features. Unless stated otherwise, our implicit models have exactly one $F$ layer (i.e., $L = 1$), in the exact form we described in §3.1. Our set of experiments is drawn from prior works where competitive implicit representation networks are evaluated on [11, 27].

Overall, our results of learning implicit representation on various domains (including images, videos, audios) suggest that the (Implicit)$^2$ approach offers clear improvements over existing explicit models used for these purposes, where we are able to achieve the same or better level of performance while being up to $3\times$ more memory-efficient and $3\times$ more time-efficient. We provide our implementation[1] and the descriptions of the tasks and datasets can be found in the appendix.

### 4.1 Image Representation

We first evaluate the difference between explicit networks and (Implicit)$^2$ networks on representing a high-resolution $512 \times 512$ grayscale image, which is a commonly used goalpost for evaluating implicit representation models from the scikit-image package [30]. In particular, we fit an implicit representation function $\Phi : (x, y) \to \mathcal{C}$, where $\mathcal{C}$ is the desired color space (and in this case $\mathcal{C} = \mathbb{R}$). We train each model for 5000 iterations (under the same setting) using all pixels in the image (i.e., batch size 262,144), and demonstrate the final peak signal-to-noise ratio (PSNR), memory consumption, and average training

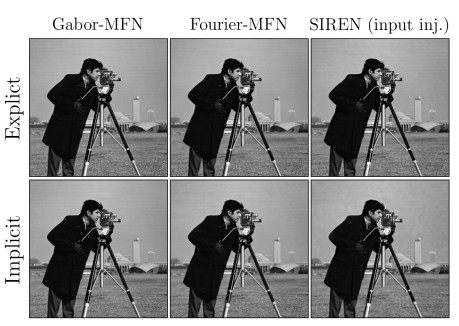

Figure 6: Learned $512 \times 512$ grayscale image

step time in Fig. 5. The results convey two interesting facts about using (Implicit)$^2$ networks: Compared to standard 4-layer explicit networks ($4L$-$256D$), even a small implicit network ($1L$-$256D$)

[1]Official implementation can be found at `https://github.com/locuslab/ImpSq`

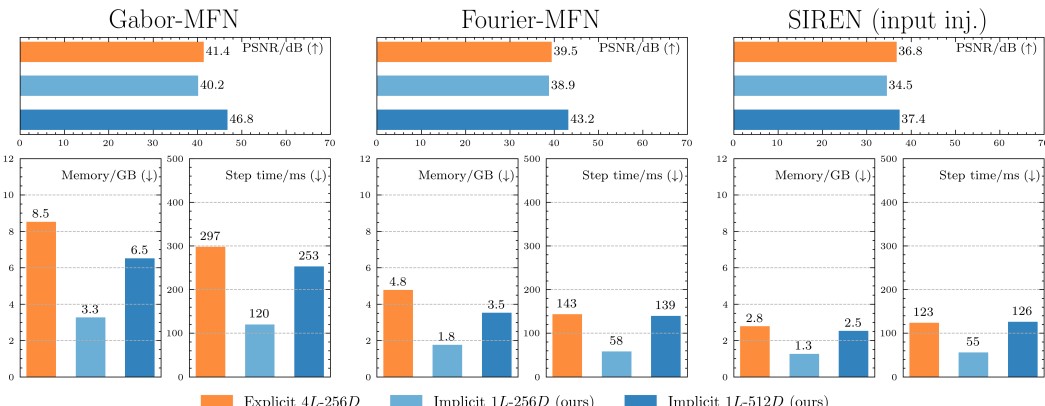

Figure 5: Comparison of step time, memory consumption and PSNR between explicit and implicit models trained on the $512 \times 512$ grayscale image.

with approximately 75% fewer parameters achieves a comparable performance, while requiring only roughly 1/3 of the training time and memory footprint. When given a higher parameter budget, an implicit network with a similar model size ($i.e., 1L\text{-}512D$) outperforms the best explicit network by a large margin.

We also demonstrate the effect of fixed-point reuse and truncated backward gradient using a 1L-512D Fourier MFN on fitting the same $512 \times 512$ image, with results shown in Fig. 3 and 4. Specifically, by reusing fixed-points in implicit layers, the solver takes significantly fewer steps to converge for the majority of the training period, making the forward evaluation of implicit layers drastically more efficient. Meanwhile, Fig. 4 shows that the $0$th-order gradient approximation in [12] (i.e. $T = 0$), albeit efficient, could substantially hurt model performance. In contrast, our proposed approximation (i.e. $T = 1$) greatly reduces the performance gap with only one additional vector-Jacobian product evaluation, thus achieving a better balance between model efficiency and effectiveness.

## 4.2  Image Generalization

In a number of applications [19, 21, 5, 8], implicit representations are used to infer unobserved parts of data. In order to demonstrate the ability for (Implicit)$^2$ networks to generalize well, we train the network on only 25% of the pixels from each image in the *Natural* and *Text* dataset, following [11], and evaluate PSNR on an unobserved 25% portion of the image. The average PSNR over all 16 high-resolution images in *Natural* and *Text* is reported in Table 1, and a visualization of the improvement by (Implicit)$^2$ models is shown in Fig. 7.

It can be seen that (Implicit)$^2$ networks improve substantially over the explicit baselines, where the best-performing implicit model outperforms the explicit counterpart by $> 0.7$ in PSNR using the same architecture. Visually, (Implicit)$^2$ networks produce a sharper and shape-consistent representation compared with the best performing explicit network. We observe similar improvements in training speed and memory consumption as in the image representation task (about $3\times$). For completeness,

| Gabor-MFN | Fourier-MFN | SIREN |
|---|---|---|

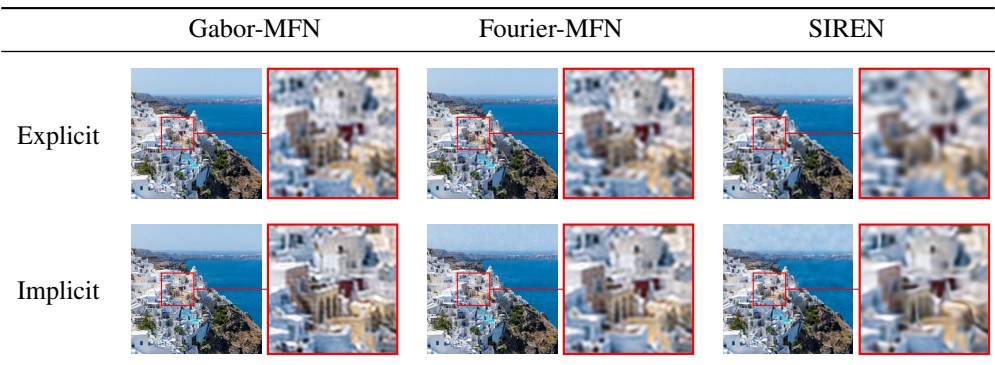

Explicit

Implicit

Figure 7: Samples of best performing explicit/implicit models learned on *Natural*

|  | *Natural* | | | *Text* | | |
|---|---|---|---|---|---|---|
|  | $1L\text{-}256D$ | $1L\text{-}512D$ | $4L\text{-}256D$ | $1L\text{-}256D$ | $1L\text{-}512D$ | $4L\text{-}256D$ |
| Fourier-MFN | $23.27 \pm 3.18$ | $23.30 \pm 3.05$ | $24.57 \pm 3.35$ | $24.64 \pm 2.11$ | $24.84 \pm 2.10$ | $26.67 \pm 2.06$ |
| Fourier-iMFN (ours) | $24.88 \pm .44$ | $\mathbf{25.19 \pm 3.64}$ | $24.52 \pm 3.33$ | $26.90 \pm 2.14$ | $\mathbf{27.19 \pm 1.83}$ | $26.48 \pm 2.04$ |
| Gabor-MFN | $24.16 \pm 3.35$ | $24.68 \pm 3.46$ | $24.65 \pm 3.38$ | $27.19 \pm 2.18$ | $27.74 \pm 2.13$ | $27.57 \pm 2.10$ |
| Gabor-iMFN (ours) | $24.91 \pm 3.41$ | $\mathbf{25.42 \pm 3.76}$ | $24.53 \pm 3.33$ | $27.53 \pm 2.18$ | $\mathbf{28.07 \pm 2.00}$ | $27.40 \pm 2.10$ |
| SIREN (input inj.) | $22.88 \pm 3.0$ | $24.52 \pm 3.28$ | $24.10 \pm 3.34$ | $24.54 \pm 2.19$ | $25.69 \pm 2.18$ | $26.21 \pm 2.19$ |
| iSIREN (ours) | $24.28 \pm 3.37$ | $\mathbf{24.92 \pm 3.58}$ | $24.05 \pm 3.39$ | $26.06 \pm 2.18$ | $\mathbf{26.81 \pm 2.09}$ | $26.31 \pm 2.20$ |

Table 1: PSNR (in dB) for all models on image generalization. The reported mean $\pm$ std is taken over the individual PSNR of the 16 images.

we also evaluate an $(\text{Implicit})^2$ network where $F$ also consists of a stack of 4 layers (rather than 1), and observe that the implicit modeling of an already deep structure does not yield much improvement.

## 4.3 Audio Representation

We further show that the proposed $(\text{Implicit})^2$ network outperforms explicit networks on representing very-high-frequency audio signals. Following [27], we train the models to fit a 7-second music piece. We observe that in all cases, the implicit modeling significantly outperforms the respective explicit models.

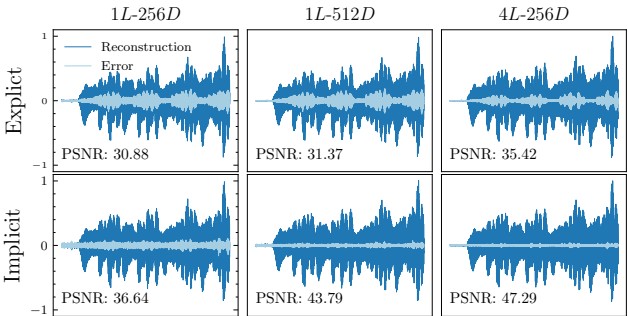

Figure 8: Audio signals represented using Fourier-MFNs. Lower error magnitudes (in light blue) are better.

In Fig. 8, we show the reconstruction results and the respective errors when using Fourier-MFN to fit the audio signal, with more results available in the appendix. Although an $(\text{Implicit})^2$ model only defines one layer, it is able to outperform explicit models in terms of PSNR by more than 23% (43.79 vs. 35.42), while still retaining the same level of memory and speed improvements as before. We also find that, in this case, having a deeper architecture in an implicit layer (implicit $4L$-$256D$) leads to improved performance.

## 4.4 Video Representation

We may add an extra dimension to the input of implicit representation function and try to learn a model for video sequences $\Phi : (t, x, y) \to \mathcal{C}$, where $t$ is the space of time. We aim to represent a 300-frame $512 \times 512$ video using each model. The PSNR results are shown in Table 2.

|  | $1L$-$1024D$ | $1L$-$2048D$ | $4L$-$1024D$ |
|---|---|---|---|
| Fourier-MFN | $24.97 \pm 1.08$ | $26.9 \pm 0.97$ | $27.64 \pm 0.90$ |
| Fourier-iMFN (ours) | $25.85 \pm 1.00$ | $27.7 \pm 0.90$ | $\mathbf{28.03 \pm 0.95}$ |
| Gabor-MFN | $26.15 \pm 1.02$ | $28.18 \pm 0.78$ | $\mathbf{29.64 \pm 0.81}$ |
| Gabor-iMFN (ours) | $26.45 \pm 0.98$ | $28.79 \pm 0.77$ | $29.20 \pm 1.04$ |
| SIREN (input inj.) | $25.12 \pm 0.96$ | $26.04 \pm 0.99$ | $26.52 \pm 0.86$ |
| iSIREN (ours) | $26.03 \pm 0.92$ | $27.08 \pm 0.93$ | $\mathbf{27.12 \pm 0.89}$ |

Table 2: PSNR for the video representation task. The reported mean $\pm$ std is taken over all frames of the video.

Unlike in the image representation setting, we observe that, due to the relatively few parameters compared to the size of the video data (the smallest model only has as many parameters as $1.3\%$ of all pixel values), model sizes have a more significant impact on the overall performance. But still, in most configurations, we found $(\text{Implicit})^2$ models lead to a consistently non-trivial improvement in performance when compared to their explicit counterparts, while doing so with equivalent or less memory budget.

## 4.5 3D Geometry Representation

To further demonstrate the benefit of $(\text{Implicit})^2$ networks over their explicit counterparts, we choose three formulations of Fourier-MFN with similar parameter count (i.e. explicit $1L$-$512D$, explicit $4L$-$256D$, and implicit $1L$-$512D$) and fitted them on several 3D object meshes with the point occupancy prediction objective similar to [18] - given input coordinate $c = (x, y, z)$, the model $\Phi : \mathbb{R}^3 \to \mathbb{R}$ is trained to predict a binary label indicating whether the point corresponding to the coordinate is located inside the target object ($0$ if the point is outside the object, and $1$ if inside). The visualized normal maps of the learned representations on one of the objects are shown in Fig. 9. We further evaluated the prediction IoU over test points densely sampled near the mesh and present them in the same figure. Experimental details and results on additional 3D models are available in the appendix.

The results show that, with a similar model size, the one-layer network modeled with the $(\text{Implicit})^2$ formulation is able to capture more details than an explicit model with an identical ($1L$-$512D$) or a deeper ($4L$-$256D$) structure, and at the same time having a training time and memory advantage similar to the ones in Fig. 5. Such an example demonstrates the superior parameter and memory efficiency of $(\text{Implicit})^2$ networks.

| Explicit
$1L$-$512D$ | Explicit
$4L$-$256D$ | Implicit
$1L$-$512D$ | Ground Truth |
| --- | --- | --- | --- |

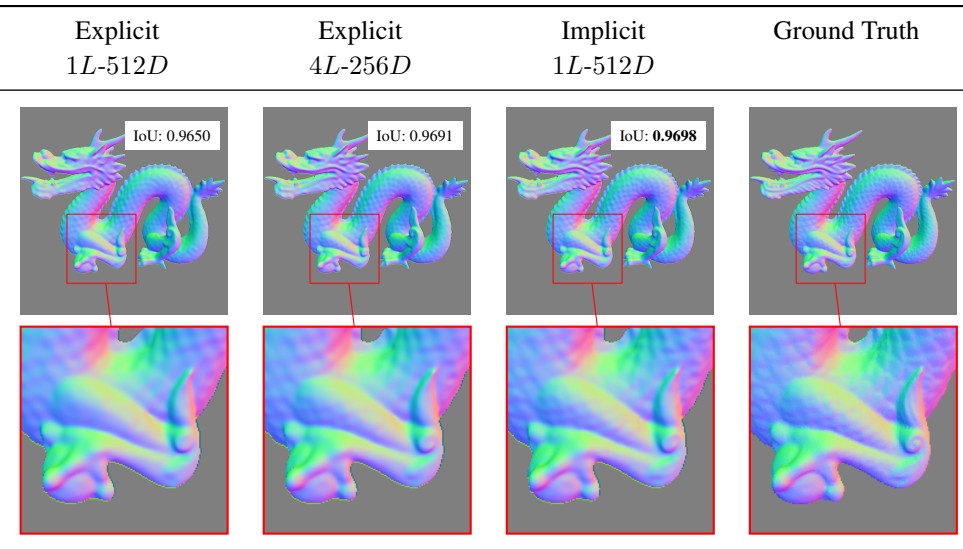

Figure 9: Normal maps and IoUs of fitted *dragon* object using Fourier-MFNs in Occupancy Network [18]. The (Implicit)$^2$ approach performs better than its deep explicit counterpart.

We note that in both the video and 3D geometry tasks, the signal is either of large dimension or defined continuously, where we may not fit the entire data into a single batch. Therefore, the implicit models usually entail more than one step to reach convergence due to the challenge in fixed-point reuse with mini-batch training (see a more detailed analysis in appendix **??**). Nevertheless, our proposed method is still able to boost the training efficiency by leveraging truncated gradient, while improving upon its explicit counterparts in most cases.

## 5    Limitations and Directions for Future Work

Our (Implicit)$^2$ approach can be significantly more time-efficient and memory-friendly than explicit networks at training time; however, at inference time, the implicit models must resort to the normal fixed-point solving. Furthermore, accelerated training in the forward pass by fixed point reuse (see §3.2) in practice can be bottlenecked by storage or hardware I/O bandwidth constraints; i.e., it works best when we can fit all or a large portion of the fixed point into memory for quick retrieval. Although such assumption can be easily satisfied for implicit representation learning in many cases, for extremely large data spaces (e.g., in the continuous domain, like Park et al. [22] and Mildenhall et al. [19], where samples are drawn continuously in $\mathbb{R}^3$; or very long video frames like in 4.4), the fixed point reuse may not always work as well and could require more solver steps.

Smarter caching strategies for the fixed points may be developed in the future, where we consider only a subset of the input space for caching throughout the training. For example, one may consider a voxel-grid-based sparse storage for the fixed points, similar to the ones used in [16], such that fixed points of any batch of inputs may be approximately obtained from interpolation of the sparse storage. We leave this direction for future work.

## 6    Conclusion

In this paper, we propose (Implicit)$^2$ networks, an application of implicit layers to implicit representation tasks. We show that these two concepts of "implicitness" complement each other well, allowing us to take advantage of their properties (e.g., fixed point reuse) to produce a significantly more efficient training routine and usually more parameter-efficient models for implicit representation learning. We demonstrate through our set of experiments that the *implicit modeling* of *implicit representations* may in many cases be conveniently used as a drop-in replacement for existing state-of-the-art explicit models like SIREN and MFN to further improve the performance and reduce the memory/computation budget required to train these tasks.

## 7 Acknowledgement

We thank Swaminathan Gurumurthy for reviewing the paper and for helpful discussions.

## 8 Funding Disclosure

Shaojie Bai was sponsored by a grant from the Bosch Center for Artificial Intelligence.

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
