# A  Proof for the Stability of iMFN and iSIREN

In this section, we show that, once we restrict $W$ in Eq. (7) and (8) in the main text to be contractive via spectral normalization, the function $F$ in both iMFN and iSIREN will also be contractive, which therefore guarantees the existence and uniqueness of a fixed point [? ]. For the remaining part of this proof, we use $\|\cdot\|_2$ on a matrix to denote its spectral norm and $\sigma(\cdot)$ to denote the set of its eigenvalues.

**Theorem 1** (Contractivity of iMFN). *Let $\|W\|_2 < 1$ and $\|g(x;\theta)\|_\infty \le 1$ for all input $x$ and parameter $\theta$. The function $F$ for the iMFN, e.g.*

$$F(z;x) = (W(z + g(x;\theta_1)) + b) \circ g(x;\theta_2) \tag{1}$$

*is contractive on $\mathcal{Z}$.*

*Proof.* Let $G_{x,\theta}$ be a diagonal matrix with $g(x;\theta)$ as the main diagonal, e.g.

$$G_{x,\theta} = \begin{bmatrix} g_1(x;\theta) & 0 & \cdots & 0 \\ 0 & g_2(x;\theta) & & \\ \vdots & & \ddots & \vdots \\ 0 & \cdots & & g_d(x;\theta) \end{bmatrix} \tag{2}$$

where $g_i(x;\theta)$ is a scalar-valued function representing the $i$-th element of $g(x;\theta)$. It is easy to see that $\|G_{x,\theta}\|_2 \le 1$ since $\max_{\lambda \in \sigma(G_{x,\theta}^\top G_{x,\theta})} |\lambda| = \max_{i \in [d]} g_i^2(x;\theta) \le 1$. Therefore, for any $z_1, z_2 \in \mathcal{Z}$, we have

$$\frac{\|F(z_1;x) - F(z_2;x)\|_2}{\|z_1 - z_2\|_2} = \frac{\|Wz_1 \circ g(x;\theta_2) - Wz_2 \circ g(x;\theta_2)\|_2}{\|z_1 - z_2\|_2} \tag{3}$$

$$= \frac{\|G_{x,\theta_2} W(z_1 - z_2)\|_2}{\|z_1 - z_2\|_2} \tag{4}$$

$$\le \frac{\|G_{x,\theta_2}\|_2 \|W\|_2 \|z_1 - z_2\|_2}{\|z_1 - z_2\|_2} \tag{5}$$

$$= \|G_{x,\theta_2}\|_2 \|W\|_2 \tag{6}$$

$$< 1 \tag{7}$$

which shows that $F$ is contractive on $\mathcal{Z}$. Additionally, the Jacobian matrix $J_F$ can be directly obtained by $J_F = G_{x,\theta}W$, which is also contractive since $\|G_{x,\theta}W\|_2 \le \|G_{x,\theta}\|_2 \|W\|_2 < 1$  □

**Theorem 2** (Contractivity of iSIREN). *Let $\|W\|_2 < 1$. The function $F$ for the iSIREN, e.g.*

$$F(z;x) = \sin(W(z + \sin(Vx)) + Ux + b) \tag{8}$$

*is contractive and has a contractive Jacobian $J_F$.*

*Proof.* Since the sine function is 1-Lipschitz, we have

$$\frac{\|F(z_1;x) - F(z_2;x)\|_2}{\|z_1 - z_2\|_2} = \frac{\|\sin(W(z_1 + \sin(Vx)) + Ux + b) - \sin(W(z_2 + \sin(Vx)) + Ux + b)\|_2}{\|z_1 - z_2\|_2} \tag{9}$$

$$\le \frac{\|W(z_1 + \sin(Vx)) + Ux + b - W(z_2 + \sin(Vx)) + Ux + b\|_2}{\|z_1 - z_2\|_2} \tag{10}$$

$$= \frac{\|W(z_1 - z_2)\|_2}{\|z_1 - z_2\|_2} \tag{11}$$

$$\le \|W\|_2 \tag{12}$$

$$< 1 \tag{13}$$

which shows the contractivity of $F$.

We now show the contractivity of the Jacobian $J_F$. Let $h = W(z + \sin(Vx)) + Ux + b$ and $D$ be a diagonal matrix with $\cos(h)$ as the main diagonal. There is $\|D\|_2 = \max_{\lambda \in \sigma(D^\top D)} |\lambda| = \max_{i \in [d]} \cos^2(h_i) \leq 1$. Therefore, the Jacobian $J_F$ is clearly contractive since it can be written explicitly as $J_F = \frac{\partial \sin(h)}{\partial h} \frac{\partial h}{\partial z} = DW$, where $\|J_F\|_2 = \|DW\|_2 \leq \|D\|_2 \|W\|_2 < 1$. $\qquad\square$

# B  Proof for §3.1

We shall proof the claim in §3.1 that the outputs of iMFN are linear combinations of the non-linear filter kernels. The proof largely follows [?], where we show that for a filter function that satisfies the *multiplicative sum* property (as defined below), every application of $F$ yields a linear combination of filter functions.

**Definition 1 (? ]).** *A filter function $g(x; \theta)$ satisfies the **multiplicative sum property** if for any input $x$ and any two parameters configurations $\theta_1$, $\theta_2$, there exists a finite set of parameter configurations $\{\overline{\theta}_i\}_{i \in [N]}$ and scalars $\{\beta_i\}_{i \in [n]}$ such that*

$$g(x; \theta_1) \circ g(x; \theta_2) = \sum_{i=0}^{N} \beta_i g(x; \overline{\theta}_i) \tag{14}$$

By A.1 and A.2 of [? ], both $g_{\text{Fourier}}$ and $g_{\text{Gabor}}$ of the original MFN satisfies the multiplicative sum property. We then have the following corollary

**Corollary 1.** *Define $z \in \mathbb{R}^d$ such that $z_k = \sum_{i=1}^{N} \beta_i^{(k)} g(x; \theta_i^{(k)})$ with parameters $\{\beta_i^{(k)}\}_{i \in [N]}$, $\{\theta_i^{(k)}\}_{i \in [N]}$ and $g(x; \theta)$ satisfies the multiplicative sum property. The output of the function*

$$F(z; x) = (W(z + g(x; \theta')) + b) \circ g(x; \theta'') \tag{15}$$

*satisfies $F_j(z; x) = \sum_{i=1}^{\overline{N}} \overline{\beta}_i^{(j)} g(x; \overline{\theta}_i^{(j)})$ with some other parameters $\{\overline{\beta}_i^{(j)}\}_{i \in [\overline{N}]}$, $\{\overline{\theta}_i^{(j)}\}_{i \in [\overline{N}]}$.*

*Proof.*

$$F_j(z; x) = \left( \sum_{k=1}^{d} W_{jk} (z_k + g(x; \theta_k')) + b_j \right) g(x; \theta_j'') \tag{16}$$

$$= \left( \sum_{k=1}^{d} W_{jk} \sum_{i=1}^{N} \beta_i^{(k)} g(x; \theta_i^{(k)}) + \sum_{k=1}^{d} W_{jk} g(x; \theta_k') + b_j \right) g(x; \theta_j'') \tag{17}$$

$$= \sum_{k=1}^{d} \sum_{i=1}^{N} W_{jk} \beta_i^{(k)} g(x; \theta_i^{(k)}) g(x; \theta_j'') + \sum_{k=1}^{d} W_{jk} g(x; \theta_k') g(x; \theta_j'') + b_j g(x; \theta_j'') \tag{18}$$

$$= \sum_{i=1}^{\overline{N}} \overline{\beta}_i g(x; \overline{\theta}_i) \tag{19}$$

$\qquad\square$

Corollary 1 states that, if every element of $z$ is a linear combination of filter functions, applying $F$ once also yields a vector consists of linear combinations of filter functions. Further, assuming the contractivity conditions in §A, the fixed point of $F$ in Eq. (15) exists and is unique. Thus, starting from $z^{[0]} = \mathbf{0}^d$, the unique fixed point $z^\star$ may be defined recursively through the fixed-point iteration with infinite steps, namely

$$z^\star = z^{[\infty]}, \qquad \text{where } z^{[t+1]} = F(z^{[t]}; x), z^{[0]} = \mathbf{0}^d \tag{20}$$

By recursively applying Corollary 1, we have $\forall t > 0, \forall k \in [d], z_k^{[t]}$ is a linear combination of filter functions, which proves the same property for $z^\star$. Since the outputs of iMFN is a linear transformation of $z^\star$, each of its elements remains a linear combination of filter functions, which finishes the proof.

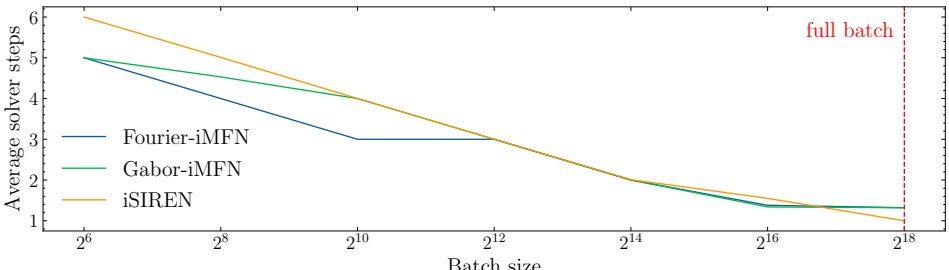

Figure 1: Average fixed-point solver steps *versus* mini-batch sizes. The average steps are taken over the last 1,000 training iterations of each model. From the figure, it can be seen that the required solver steps is inversely related to the batch size. However, only when the batch size is sufficiently large ($> 2^{16}$) can the majority of the fixed-points converge in 1 step (average solver steps $< 1.5$)
.

## C  (**Implicit**)$^2$ **Model and SGD**

The explicit models we compared to in this work (except ones for video and 3D occupancy representations) are trained with full batch gradient descent. Given the success of stochastic gradient descent (SGD) in deep learning, it is natural to wonder whether training the explicit models in mini-batches would yield better training performance or generalization. Empirically, we find the answer to be *no*, where we evaluated different mini-batch sizes {64, 256, 1024, 4096} in the image generalization task and, given the same compute budget, find the resulting models to still be inferior to (Implicit)$^2$ networks both in terms of training and testing error.

Another natural question to ask is whether (Implicit)$^2$ models, especially the fixed-point reuse component, can be applied in SGD training as well, since SGD is often more feasible in general machine learning tasks. To explore this, we maintain a fixed-point cache for each training sample (i.e. a coordinate) in the image fitting task, and experiment with different batch sizes at which the fixed-points are computed by running the ordinary fixed-point solving routine. We cached the fixed-points at every training iteration and reuse them whenever a sample is revisited. As the training error stabilizes, the number of steps the solver takes to converge from the cached fixed points is reported in Fig. 1.

It is clear from the table that replacing the fixed point iteration with a single forward iteration (which is required to match the efficiency of the explicit models) is only feasible when the batch consists of a significant portion of the training data. Therefore, the proposed replacement of the fixed-point solver, in general, can only work with data of relative low dimensions, such as images coordinates (which is extensively studied in this paper). Nevertheless, it can also be observed that the fixed-point cost saving is proportional to the mini-batch size, which may motivate fixed-point reuse in DEQ trained on other tasks as a general strategy to save computation. A potentially more interesting future direction, as covered in §**??**, is to leverage some spatial-temporal priors of the data (e.g. spatially close points in 3D space or consecutive frames in a video stream) to further reduce the storage and updates needed to maintain the cache.

## D  **Training Specifications**

In this section, we provide detailed training specifications and hyperparameter choices used to produce our main results. Unless specified otherwise, we apply spectral normalization as in [**?** ] on all $W$ and used an Adam optimizer with ($\beta_1 = 0.9, \beta_2 = 0.999, \epsilon = $ 1e-8) for training.

### D.1  **Image Representation**

**Data.**  The $512 \times 512$ grayscale image is imported from the scikit-image package [**?** ] of version 0.16.2 via skimage.data.camera(). We notice a recent change to this image in the latest scikit-image package due to copyright issues, and therefore call attention to the readers to use the correct data.

**Initialization.** We apply a scaling of 256 to all input layers and choose $\alpha = 3.0$ for the initialization of the Gabor filters

**Training Details.** We use all (262,144) pixels in each batch and train the models for $5,000$ steps with a constant learning rate 1e-3.

**Hardware.** The models are trained and evaluated using a 11GB NVIDIA GTX 1080TI GPU.

**Additional Note.** Figure 3 and 4 in the main text are generated using a 1L-512D Fourier-iMFN with the same data and hyperparameters, except that we control whether the fixed points are reused and the number of Jacobian evaluations $T$ during the backward pass, respectively.

### D.2 Image Generalization

**Data.** The *Natural* and *Text* dataset we use in this experiment was originally created by [? ] and consists of 16 natural or word overlay images of resolution $512 \times 512$. The data is made available alongside the code.

**Initialization.** We apply a scaling of 128 to all input layers and choose $\alpha = 3.0$ for the initialization of the Gabor filters

**Training Details.** Following prior works like [? ], we train the networks on 25% of the data and evaluate on a disjoint 25% set (a visualization is shown in Fig. 2). For every model, we train on each image for $2,000$ steps with learning rate $\eta = $ 1e-3.

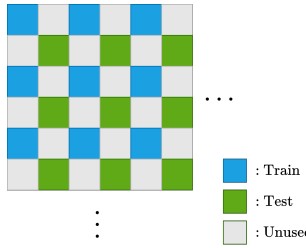

Figure 2: Train/Test split in the image generalization task

**Hardware.** The models are trained and evaluated using a 11GB NVIDIA GTX 1080TI GPU.

### D.3 Audio Representation

**Data.** Following [? ], the audio signal is retrieved from the first 7-seconds of Bach's Cello Suite No. 1: Prelude and is also made available in the submitted code.

**Initialization.** To account for the high-frequency nature of the audio signals, we applied a scaling of 25,000 to the input linear layers. We empirically find that implicit models without spectral norm yields better results while remains stable throughout training. Therefore, in the audio experiments, we apply weight normalization on $W$ instead of spectral normalization.

**Training Details.** We train the models using the entire audio piece in each batch (of size $308, 980$) for $3,000$ steps with a constant learning rate 1e-3.

**Hardware.** The models are trained and evaluated using a 11GB NVIDIA RTX 2080TI GPU.

### D.4 Video Representation

**Data.** The full video clip used in the video representation task is publicly available at

https://www.pexels.com/video/the-full-facial-features-of-a-pet-cat-3040808

We choose a $512 \times 512$ center crop of this video for training and evaluation, following [? ? ]. The data is also included as part of our submitted code.

**Initialization.** We apply the same input scaling to both explicit and implicit models with the same base architectures according to Table 1. The individual scaling factors were chosen to optimize the performance of the explicit networks. Additionally, we choose $\alpha = 6.0$ in the 4-layer models and $\alpha = 3.0$ in the 1-layer models for the initialization of the Gabor filters. We apply weight normalization instead of spectral normalization on all $W$.

**Training Details.** We train each model for $10,000$ steps with a constant learning rate 1e-3 and a batch size of 50,000. The batch size is chosen such that the largest explicit model (i.e. Gabor-MFN-$4L$-$1024D$) is able to fit in the GPU memory.

|  | $1L$-$1024D$ | $1L$-$2048D$ | $4L$-$1024D$ |
|---|---|---|---|
| Gabor-MFN | 128 | 256 | 256 |
| Fourier-MFN | 128 | 256 | 256 |
| SIREN (input inj.) | 128 | 128 | 256 |

Table 1: Scaling factors for explicit/implicit models

**Hardware.** The models are trained and evaluated using a 11GB NVIDIA RTX 2080TI GPU.

### D.5 3D Model Occupancy Representation

**Data.** The 3D objects (i.e. *dragon*, *buddha*, *armadillo*, *lucy*) used in this experiment are retrieved from the Stanford 3D Scanning Repository.

**Initialization.** We apply a scaling of 256 to all input layers.

**Training/Evaluation Details.** Our training and evaluation procedure largely follows **?** ]. The 3D object meshes are scaled and transformed to fit in side the unit cube $[0, 1]^3$. We use the point-in-mesh algorithm in the Trimesh library to efficiently obtain the ground truth point occupancy label for the input coordinates. Each model is trained on the binary cross-entropy for $10,000$ steps with a *constant* learning rate 5e-4. At inference, the depth maps are generated based on the first ray intersection to the isosurface $\Phi(x, y, z) = 0.5$ from the camera, and the normal maps (as shown in Fig. **??**) are computed using derivatives of the depth maps. The test points used to compute IoU are sampled from *i.i.d.* Gaussians, each with a mean at uniformly random point on the mesh and a standard deviation of 0.01.

**Hardware.** The models are trained and evaluated using a 11GB NVIDIA RTX 2080TI GPU.

## E  Additional Visualizations & Experiments

### E.1  Image fitting on CelebA

We evaluated both explicit and implicit models on fitting each of the first 100 images in the CelebA dataset [**?** ]. The images were center-cropped and rescaled to $128 \times 128$. We used a learning rate of 3e-3 and trained each network for 3,000 steps. The results are presented in Table 2. This experiment shows that the (Implicit)$^2$ networks still outperform explicit counterparts on data of larger scales, further demonstrating their effectiveness.

|  | $1L$-$128D$ | $1L$-$256D$ | $4L$-$128D$ |
|---|---|---|---|
| Fourier-MFN | $35.48 \pm 4.01$ | $36.41 \pm 4.30$ | $44.27 \pm 2.75$ |
| Fourier-iMFN | $38.03 \pm 2.78$ | $\mathbf{46.70 \pm 2.78}$ | $44.72 \pm 2.45$ |
| Gabor-MFN | $39.08 \pm 3.52$ | $45.85 \pm 3.70$ | $42.55 \pm 2.89$ |
| Gabor-iMFN | $39.39 \pm 2.83$ | $\mathbf{47.72 \pm 2.02}$ | $41.96 \pm 3.17$ |
| SIREN | $34.41 \pm 3.70$ | $36.97 \pm 3.98$ | $38.78 \pm 4.06$ |
| iSIREN | $36.75 \pm 3.62$ | $\mathbf{40.90 \pm 3.08}$ | $39.40 \pm 4.07$ |

Table 2: PSNR for image fitting on CelebA

### E.2  Gabor-MFN and SIREN for Audio Representation

As promised in §4.3 of the main text, we show the results of the remaining two models in the audio representation tasks in Fig 3 and 4. We observe that implicit Gabor-MFN outperforms its explicit counterparts, while iSIREN performs slightly worse. We hypothesize the marginally inferior performance of iSIREN is likely due to the absence of spectral normalization (as mentioned in Sec

D.3), which makes implicit models less stable toward the end of training. Empirically, iSIREN suffers the most from this instability which thus causes the relatively weaker performance. However, even with such instability, the best implicit models still perform competitively to the best explicit models, while have a better memory/training time profile due to implicit modeling with fixed-point reuse and truncated backward pass.

### E.3   Additional Visualizations for Image Generalization

We show additional outputs of learned implicit representation models in the image generalization task (Fig. 5 and Fig. 6). The models we compare (implicit $1L$-$512D$ and explicit $4L$-$256D$) have a similar parameter count, while the implicit network in general requires less memory and training time due to the fixed point reuse and truncated backward gradient. For all types of networks (Gabor-MFN, Fourier-MFN, and SIREN), the $(\text{Implicit})^2$ formulation consistently outperform the explicit counterparts in visual quality, while also enjoys more computational efficiency.

### E.4   Additional Visualizations for 3D Object Occupancy

We show the visualized normal maps for 3 additional 3D objects from Stanford 3D Scanning Repository, i.e. *buddha*, *armadillo*, and *lucy*, in Fig. 7. The results show that the shallow implicit network ($1L$-$512D$) yields reconstructions with details on par or better than the deep explicit alternative ($4L$-$256D$) and significantly outperforms the shallow explicit network ($1L$-$512D$).

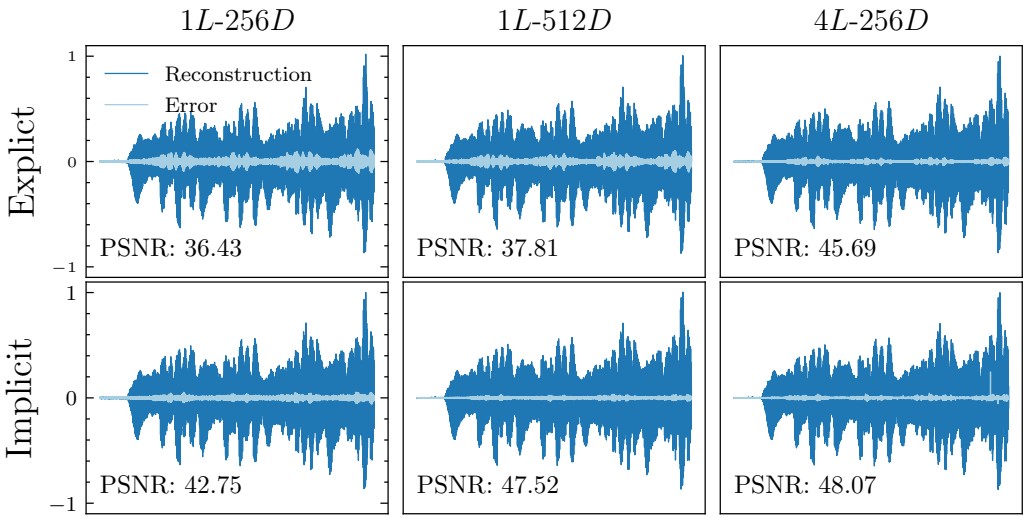

Figure 3: Audio representation task using explicit/implicit Gabor-MFN

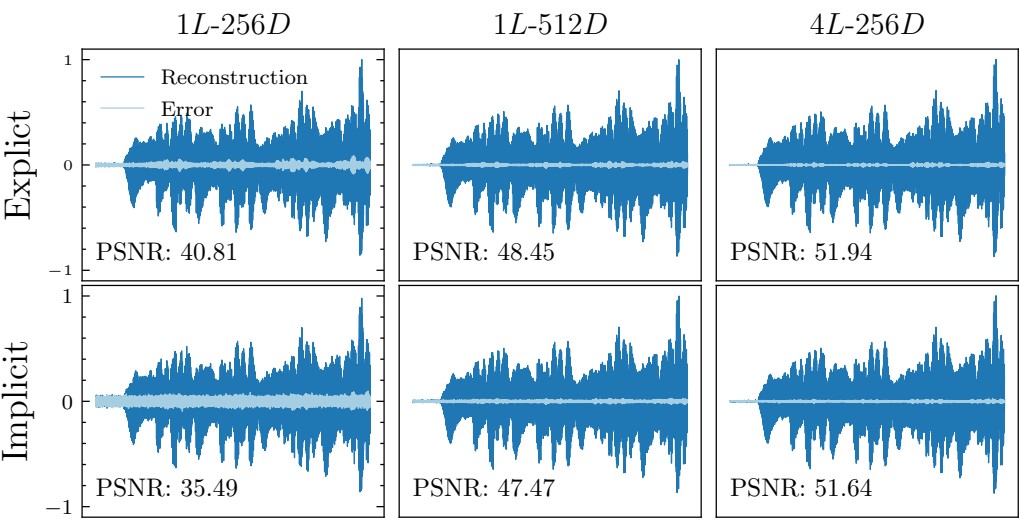

Figure 4: Audio representation task using SIREN (input inj.) or iSIREN

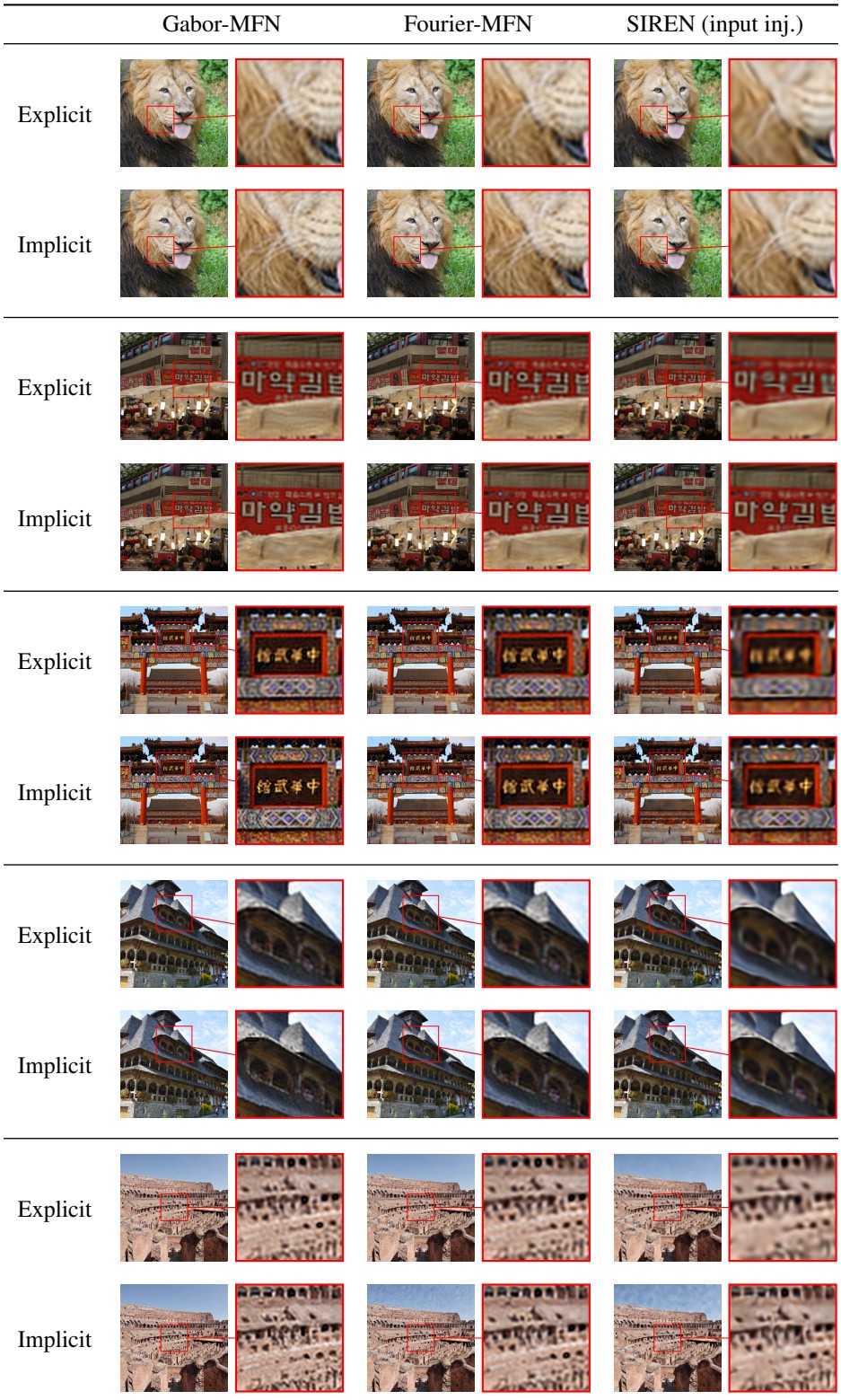

Figure 5: Implicit $1L$-$512D$ *vs.* Explicit $4L$-$256D$ on *Natural*

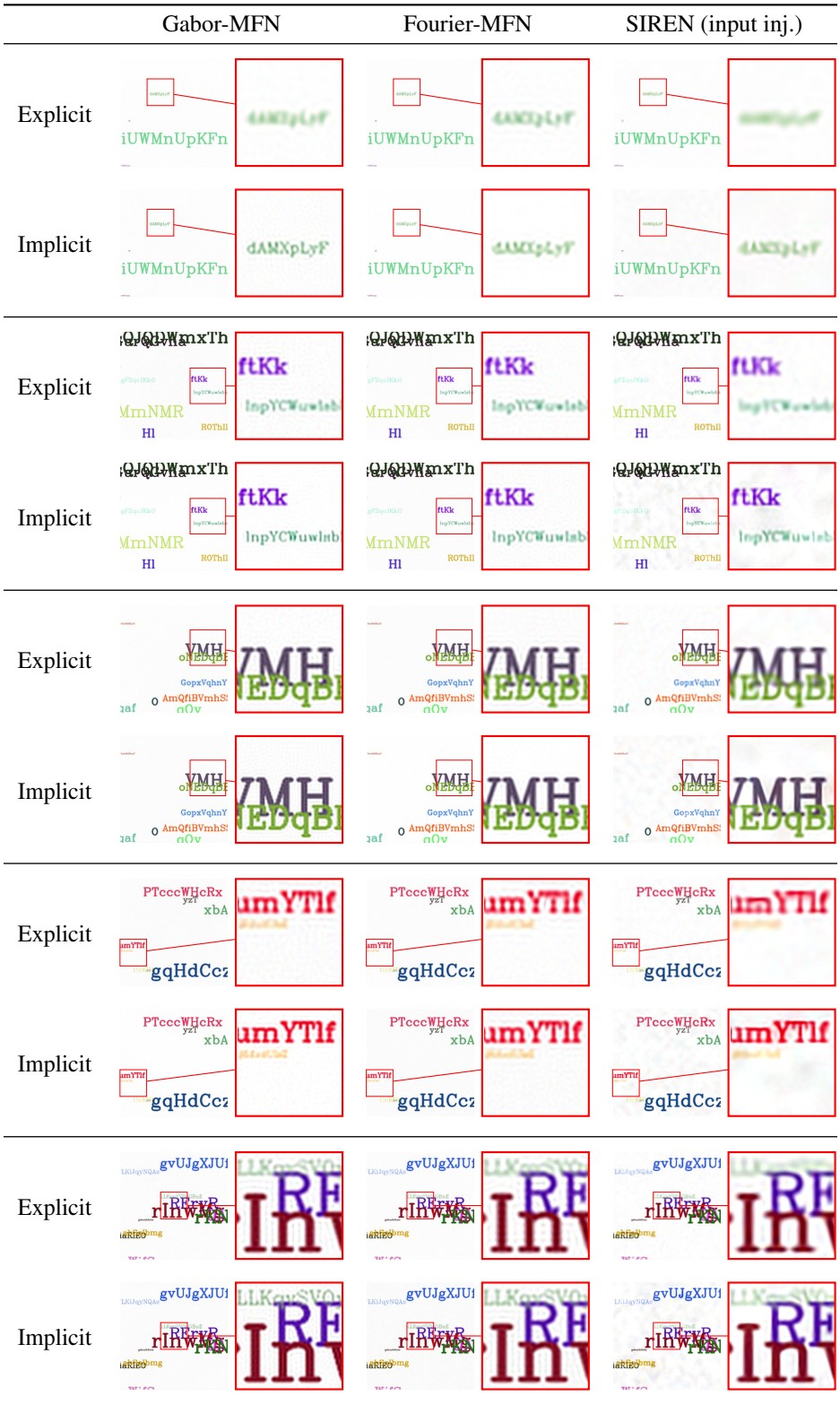

Figure 6: Implicit $1L$-$512D$ *vs.* Explicit $4L$-$256D$ on *Text*

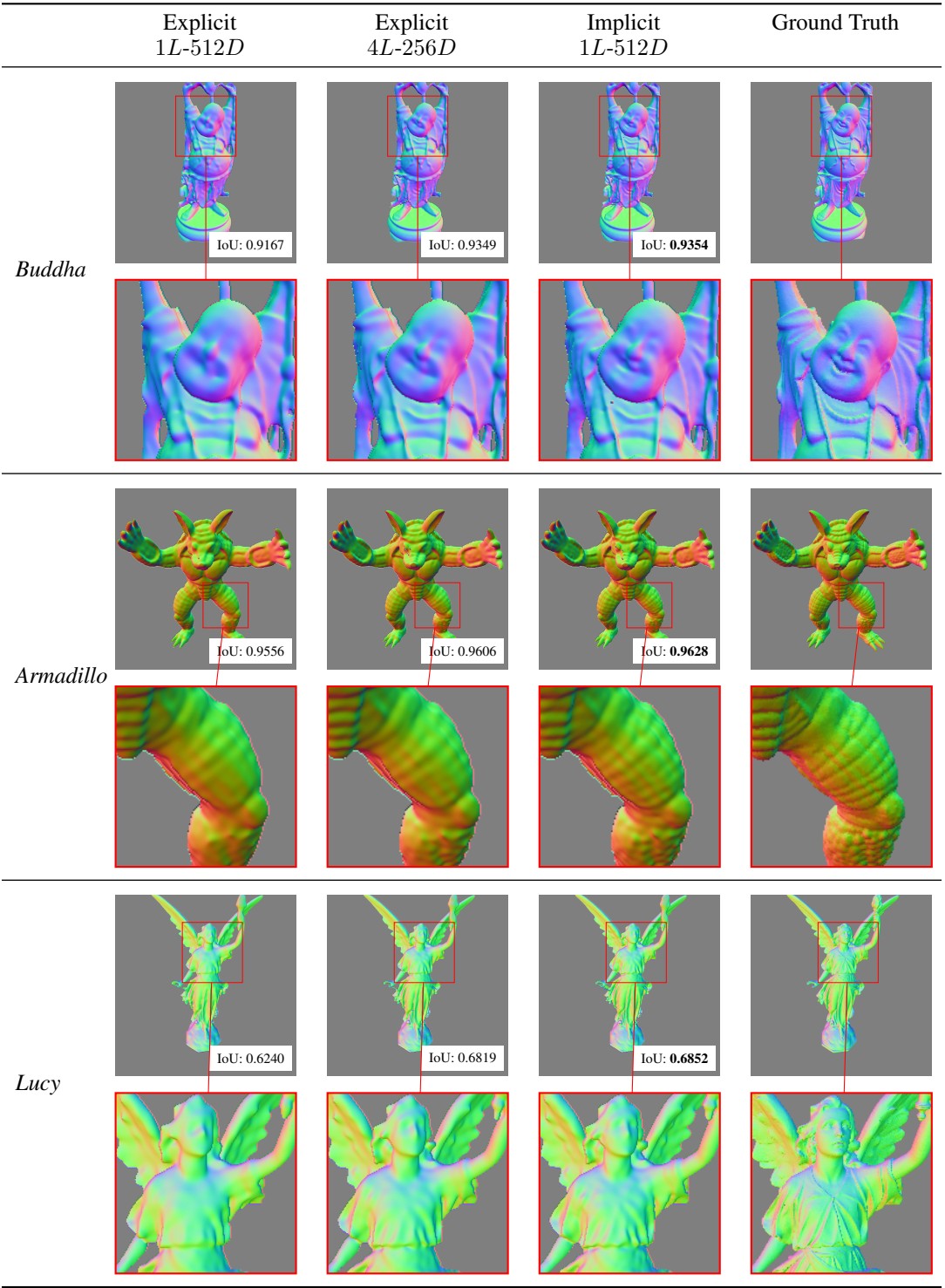

Figure 7: Additional results on fitting objects from Stanford 3D Scanning Repository