# OpenReview forum: "$(\textrm{Implicit})^2$: Implicit Layers for Implicit Representations"
_NeurIPS.cc/2021/Conference — NeurIPS 2021 Poster_

### Official Review · Reviewer_6HGN · 2021-07-05

**Rating:** 6
**Confidence:** 5

**Summary:**

This work suggests that combining implicit-depth models (e.g. DEQs) with implicit representation approaches (e.g. SIREN) can bring performance improvements. The method is evaluated on image, audio and video representation.

**Limitations And Societal Impact:**

The authors provide a satisfactory discussion on limitations.

**Main Review:**

This work aims to develop several ideas simultaneously and does not always achieve the prescribed goal. At its core, this paper is about combining implicit-depth models with implicit representations to improve empirical performance in data representation tasks. The second major component of this paper is an attempt at introducing methodological improvements to implicit-depth models (particularly DEQs [1]) with root finding solution reuse and truncated gradients.

The work does a really poor job at motivating the combination of these two models (beyond the similarity in nomenclature), or performing an analysis on the reasons why performance improvements should even be expected. The novel material (page 5 onwards) introduces a few changes that can be applied to all DEQ models. These have nothing to do with implicit representations, yet in this work they are sold as a core component of the model. See for example Figure 1, where (Implicit)$^2$ is exactly a DEQ  with the proposed improvements.

The proposed improvements to implicit models are reasonable. Root reutilization is definitely useful and is simultaneously being proposed in other recent papers on the topic. I would also like to point out that the original DEQ implementation also features root solution reuse (see for example [2]), so this section not particularly surprising. Truncated gradients and spectral regularization are similarly natural improvements over the DEQ formulation which I was glad to see here. I would suggest the authors decouple these improvements with their combination of models, perhaps introducing them beforehand.

To me, the greatest limitation of this work are the experiments. Given that the work does not appear motivated by theory or particular insights, I expect strong results to carry the work. I'm seeing marginal improvements, with the most impressive being SNR of smaller implicit Gabor/Fourier-MFN compared to larger explicit variants. Implicit SIRENs actually seem to perform worse in several tasks, see for example audio reconstruction in the supplementary (Figure 3). Image and 3D model representations do appear slightly sharper. The code submission is organized well.

Overall, I believe this paper is borderline, and I am definitely willing to modify my score, particularly if the authors can provide a stronger motivation behind the merging of these two modeling frameworks. As it stands, I see this work only as a preliminary exploration which might spur more definitive follow-up work on better implicit representation models.


* (1) Deep Equilibrium Models
* (2) https://github.com/locuslab/deq/blob/beta/DEQModel/models/transformers/deq_transformer.py
* (3) Stabilizing Equilibrium Models by Jacobian Regularization

-----------------

Minor:

> While the modification seems simple, the addition of implicitness in fact yields a much richer set of functions than the ones can be represented by a forward pass of F alone

This is wrong as a general statement and is not proven anywhere in the paper. Hand-wavy general comments such as this one have no place in the paper.

> Therefore, the forward and backward passes of implicit models can rely on independent black-box solvers, and these models consume only a constant training memory (as they do not need to store anything)

Incorrect and misleading. Implicit models (both Neural ODEs and ) require storage of various quantities to ensure low errors. Backsolve continuous adjoints for Neural ODEs can give you gradients at no additional memory cost, but can also introduce unbounded errors depending on the solver. Similarly, DEQs require storage of root solutions for faster training. Repeating the fixed point iteration at each training step, while free of memory overheads, is often too slow. There are also other costs discussed in [3] (3.4)

**Time Spent Reviewing:**

12

---

> ### Author Response · Authors · 2021-08-10
> **Response to Reviewer 6HGN**
>
> We sincerely thank the reviewer for their valuable feedback. We did attempt, e.g. in lines 34-52 in the introduction, to lay out our case for the synergies between the two types of implicit models: namely that a commonly applied implicit representation learning routine (specifically, full-batch training) can help mitigate the computational challenges of typical implicit models, and implicit models can make INRs truly *implicit* which ideally brings better optimization properties similar to what observed in [1] (and confirmed by our experiments, e.g., memory efficiency).  However, we will attempt to further emphasize this point, with the additional comments below. We also see a few discrepancies between the reviewer’s comments and the previous literature as well as the results we obtain. Below we try to respond to every issue brought up by the reviewer and hope to clear some of the confusion.
>
> #### **1. Motivation to combine implicit modeling and implicit representations**
>
> We would like to first clarify that (at least by the time of the submission) we were not aware of other works that have proposed fixed-point/root reuse prior to this work. We are very familiar with the code mentioned by the reviewer, and would like to clarify that it does **not** implement root solution reuse, and instead sets the initial guess to 0 (see https://github.com/locuslab/deq/blob/master/DEQ-Sequence/models/deq_transformer.py#L338). In fact, DEQ [1] was primarily used for large-scale tasks (e.g., language modeling) that can only be trained with many mini-batches, which prohibits the very notion of fixed point reuse (e.g., the model will have been updated by an undesirably large number of steps before a fixed point is poorly reused; there are also storage issues). In contrast, in INR tasks where typically we train in full-batch mode or with one mini-batch constituting a significant portion of the data, as we show in this paper, we can train with **significantly better efficiency** and converge with as few as 1 solver step.
>
> In the following table, we present the average required # of solver steps to converge to a given error level (1e-2) when the fixed point is evaluated and cached with different batch sizes. (Stats reported from the last 1000 training iterations of the image representation task).
>
> | Model type \ Batch size |  64  |  256 | 1024 | 4096 | 16384 | 65536 | 262144 (full batch) |
> |-------------------------|:----:|:----:|:----:|:----:|:-----:|:-----:|:-------------------:|
> | Gabor-iMFN              | 5.00 | 4.00 | 3.00 | 3.00 |  2.00 |  1.38 |         1.32        |
> | Fourier-iMFN            | 5.00 | 4.53 | 4.00 | 3.00 |  2.00 |  1.34 |         1.32        |
> | iSIREN                  | 6.00 | 5.01 | 4.00 | 3.00 |  2.01 |  1.55 |         1.00        |
>
>
> It is clear from the table that, only when each batch consists of a large proportion of the data, replacing the solver with one forward iteration based on the previous cached fixed point makes sense. Additionally, if the size of the training data is large (e.g. the language modeling task in [1]), simply maintaining the cached fixed points can pose a significant storage challenge. While full-batch (or very-large-batch) training is rarely applicable in most machine learning tasks, INR is one of the few applications that commonly use it, especially on fitting relatively small data such as images and audio. Therefore, our motivation to combine the two is natural and introduces benefits to both sides.
>
> #### **2. Concerns about additional storage induced by solvers.**
>
> We totally agree with the reviewer that in most conventional implicit models (specifically DEQ), additional quantities often need to be stored to ensure low solver error (e.g. the iteratively updated low-rank preconditioning terms when one uses Broyden’s method). However, since this work is meant to motivate both efficient and effective INR using implicit models, we find that using advanced solvers with stronger convergence guarantees often induces overwhelming computational/memory overheads. This motivates the use of a naive fixed point iteration scheme (along with spectral regularization, see Sec. 3.2), whose solving process actually introduces no additional memory cost at all (in contrast to the “hidden solver costs” that more advanced solvers could suffer from, see [2]).
>
> Regarding the extra memory cost that may be induced by storing root solutions, we would like to point out that the cached root solution 1) does not require gradient; and 2) interacts with the mapped input g(x) through an addition in our formulation. Therefore, using the cached root solution does not necessarily induce extra memory cost since we perform the addition inplace. It can also be seen from Fig. 5 that in our Pytorch implementation, the memory cost of training an (implicit)^2 model is equivalent to that of the explicit model with identical base architecture. However, we agree that the claim “do not need to store anything” in the paper is too assertive and does not apply to all implicit models. We are very happy to follow the reviewer’s advice and state it more accurately in the final version of our paper.
>
> #### **3. The significance of improvements over explicit models**
>
> Indeed, for (implicit)^2 models, the improvements over explicit models in terms of fitting error (PSNR)  in some cases may not be very significant, especially when the base architecture is already reasonably deep (e.g. comparing 4L implicit/explicit networks). However, as the reviewer points out, in the majority of presented experiments, a shallow (e.g. 1L-256D) (implicit)^2 network can reach competitive performance compared with a deep (e.g. 4L-256D) explicit network, while enjoys significantly reduced memory and computational cost. At the same time, it is particularly interesting if the number of parameters is taken into account, since a 1L-256D implicit model uses ~75% less parameters than a 4L-256D explicit model. Through the (implicit)^2 network, we obtained a significantly more parameter-efficient INR model, which can be of special interest in the data compression applications of INR [3].
>
> We hope these address the questions/concerns that the reviewer has, and that the reviewer could consider adjusting the score accordingly. Please let us know if there’s anything else that we can clarify.
>
> References:
>
> [1] Bai et al. Deep Equilibrium Models.
>
> [2] Bai et al. Stabilizing Equilibrium Models by Jacobian Regularization.
>
> [3] Dupont et al. COIN: COmpression with Implicit Neural representations.

---

> > ### Comment · Reviewer_6HGN · 2021-08-29
> > **Response to the authors**
> >
> > Apologies for the late response. Thank you for the detailed rebuttal. After evaluating all responses and in particular concerns raised by other reviewers, I have decided to raise my score to a 6. The work still has weaknesses, particularly related to the experimental evaluation, but it might still be beneficial to get these preliminary results out there given that the idea itself has value.
> >
> > Please remember to incorporate suggested feedback and changes. I would also note that it might be a good idea to scan the literature for concurrent applications of solution reuse so that interested readers can easily find other applications of DEQs and "implicit" networks.

---

### Official Review · Reviewer_R9UM · 2021-07-15

**Rating:** 7
**Confidence:** 3

**Summary:**

The paper proposed $($Implicit$)^2$, which replaces the internal layer of implicit neural representation (INR) with implicit layers. The proposed method is efficient in terms of memory and training time. For the fast training, the author proposed the re-usage of fixed-point and also used truncated backpropagation (approximated gradient). The method is sensible and also empirically demonstrates its effectiveness.

**Ethical Concerns:**

No ethical concerns found.

**Limitations And Societal Impact:**

The authors have addressed the limitations.\
For societal impacts, I agree that this work may not have negative impacts.


**Main Review:**

-----------------------------------------------------------------
**Strength**
1. The motivation of using an implicit layer into implicit neural representation (INR) makes sense (as it significantly reduces memory utilization).

2. The performance gains (for both efficiency and training time) are significant.

3. Although there are no theoretical guarantees, the authors empirically showed that the re-usage of fixed-point and truncated back gradients are sensible.

4. The writing and presentation are clear.
-----------------------------------------------------------------
**Weakness**

**1. Some scalability concerns: the experiments are demonstrated under somewhat small-scale datasets**
- If possible, can the author show experiments requiring a large number of INR layers, i.e., a complex signal? For instance, 3D shapeNet needs eight layers to be represented with a Fourier feature network [2]. Demonstrating experiments under such conditions will be interesting for $($Implicit$)^2$.
- Alternatively, datasets that are used in Neural Radiance Fields [1] will be significant.
I believe such experiments are the mainstream task of INR.

**2. (Minor) Comparison of inference time**
- I recognize that direct comparison of inference time is somewhat difficult (due to CPU utilization for the solver). However, still believe the paper should include such part.
- If possible, can the author compare the inference time of $($Implicit$)^2$ with the baselines?

**3. (Minor) The variance seems to be big in Table 1**
- Can the author evaluate the images for a large number than 16, e.g., 250 samples, by following [3]? (Of course, the author can use a smaller dataset, e.g., CelebA)
-----------------------------------------------------------------
**Justification of the rating**\
I recommend accept. I believe the idea of the implicit layer is well suited for INR and also shows significant efficiency. I do enjoy reading the manuscript.

I still have some concerns about the scalability issues; therefore, I politely request the authors to share the experimental results mentioned in the weakness part.

-----------------------------------------------------------------
**Reference**\
[1] NeRF: Representing Scenes as Neural Radiance Fields for View Synthesis. Mildenhall et al., ECCV 2020\
[2] Fourier Features Let Networks Learn High Frequency Functions in Low Dimensional Domains. Tanick et al., NeurIPS 2020\
[3] Learned Initializations for Optimizing Coordinate-Based Neural Representations. Tanick et al., CVPR 2021




**Time Spent Reviewing:**

8

---

> ### Author Response · Authors · 2021-08-10
> **Response to Reviewer R9UM**
>
> We thank the reviewer for their thoughtful review and for appreciating our work. We hope the following response would provide further insights into the effectiveness of (Implicit)^2 models compared with the explicit modeling of INR.
>
> #### **1. Experiments requiring a large number of INR layers (i.e., a complex signal)**
> On the final page of the supplement, we presented results on fitting explicit/implicit networks on the occupancy function of four different 3d models. These are also complex signals which were originally modeled by an 8-layer FFN. Our results show that, although we still employed networks with a maximum depth of 4 for the explicit models, that even a one-layer (Implicit)^2 network outperforms its deep, explicit variant.
>
> We also agree with the reviewer that the inclusion of NeRF could potentially add value to this work. However, it is generally not possible to perform full-batch training when modeling these complex signals (e.g. long videos, 3d objects, and radiance fields) with the implicit model due to the size of the data. This forbids fixed-point reuse and forces expensive forward passes, which we have briefly mentioned in the limitation section. Since fixed-point reuse is a central part of a computationally efficient implicit model, within the scope of this work we wish to mainly focus on small data that can be trained in full-batch, such as images and audio signals, while we acknowledge the potential of extending (implicit)^2 networks to more complex domains in INR such as neural rendering and wish to do so in follow-up works.
>
> #### **2. Inference time comparison of (implicit)^2 with the baselines?**
> For a general DEQ model, the most commonly used solvers (e.g. Broyden’s method, Anderson acceleration) can be implemented fully using GPU operations (see the implementation of [1]). In the scope of this paper, to maximize the efficiency improvement, we use a simple fixed-point iteration as the solver for the (implicit)^2 model, where each step costs approximately the same as the evaluation of one layer in explicit networks. At inference, although we cannot reuse the fixed point and need to iteratively solve for the fixed point, empirically we are able to do so with at most 6 solver steps, thanks to spectral regularization (see Sec. 3.2; which regularizes the implicit model stability). Therefore, the (implicit)^2 models we evaluate have a runtime efficiency comparable with their deep (4L) explicit counterparts. We further note that implicit models without spectral regularization are often unstable under the naive fixed-point iteration and may incur >30 solver steps for convergence.
>
> #### **3. Experiments with more samples on CelebA**
> We present the results for an additional experiment on 100 128x128 CelebA images in section 2 of the general response. The results agree with the main argument of our paper: (implicit)^2 models produce a boost in performance compared to deep explicit models while using less memory/computation. Although the size of the dataset is less than what the reviewer proposed, we are happy to integrate further results on larger datasets in our final version.
>
> [1] Bai et al. Deep Equilibrium Models.

---

> > ### Comment · Reviewer_R9UM · 2021-08-27
> > **Response**
> >
> > I deeply apologize for the late reply and thank you for the detailed response.
> >
> > I would like to keep my score as the experimental result seems promising. However, if possible, please add a quantitative result of the 3D object result in the final draft along with the qualitative results.

---

### Official Review · Reviewer_7RDY · 2021-07-16

**Rating:** 6
**Confidence:** 3

**Summary:**

Summary
------------------
This manuscript proposes the use of implicit differentiation (i.e. using the implicit function theorem)
to train variants of so-called "implicit" representations (also known as "coordinate based networks")
the authors develop.
Specifically the manuscript extends the formulations of SIRENs and MFNs in the form of Deep Equilibrium
Networks (DEQ) that can thus be trained through implicit differentiation. To make training of those
architectures efficient, the manuscript suggests two approximations. The first approximation lies in the
forward pass and circumvents the use of an expensive Fixed Point Solver. The second approximation
lies in the backward pass and circumvents the expensive inversion of typically "large" Jacobian matrices.
The authors evaluate their new architectures and training schemes on image, audio and video fitting and
show that smaller Implicit^2 networks can fit signals better than SIREN or MFNs.

**Limitations And Societal Impact:**

I am not sure this is applicable to this paper.

**Main Review:**

Main review
------------------
I find the A+B idea of mixing two seemingly orthogonal concepts (implicit representations and implicit
representations) interesting. The writing is globally clear, even though the style is rather
heterogeneous through the manuscript. Some sections would definitely deserve some more love (and clarity).

The motivation of the paper is also clear to me: we want better and more memory friendly "implicit
representations" and the use of implicit differentiation techniques that are typically cheaper
(e.g. in Neural ODEs with the method of the adjoint) seems promising!

Despite those interesting ideas, and the promise of more memory efficient implicit representations (at the
cost of more time-consuming forward and backward passes), the paper fails at convincing me this is the case.
In fact, if I understand correctly Figure 5, a glance at it shows immediately that there is no gain in the
memory consumption between the implicit differentiation method proposed by the authors and the "explicit"
differentiations of conventional SIRENs and MFN. Furthermore, implicit differentiation is indeed slower. All
in all, we end up with a slower method that is as memory consuming. Therefore, I would absolutely requalify
the claims in the introduction.

As a matter of fact, the authors end up twisting those initial claims (that turn out to be misleading)
through the paper, and we arrive at Section 4 where essentially the main claim is that with a smaller
number of parameters (a smaller architecture) the architecture and implicit differentiation scheme
proposed by the authors seem to perform better than a SIREN or MFN. As a summary, Implicit^2 has the
same memory consumption for an equivalent architecture as a conventional SIREN or MFN, runs slower,
but seems to perform better! Consequently, if the paper is accepted, the storyline must be reworked,
the motivation on why implicit differentiation helps to do so too, and hints or experiments on why
those architectures perform better should be provided.

Originality and significance
----------------------------------------
I find this work interesting, I am not completely sure about its level of significance but believe significance
would be increased if code for implementing and training this architecture was distributed. In terms of
originality, this work falls in the category of A+B papers, combining two existing ideas, it seems relatively
well executed: results despite being on toy problems seem promising.

Questions
------------
- Can the authors explain why the implicit^2 architecture ends up consuming as much memory as
SIREN or MFN for a same architecture contrary to what was expected and motivated the whole paper?
- Can the authors explain/hint why this is the case that the SIREN or MFN perform worse than
Implicit^2 on the tasks shown in the paper? Is there some inductive bias in Implicit^2 that helps
for the task presented? Is it simply that those optimize better?


More specific comments and typos
-------------------------------------
- I would suggest to improve the caption of all the figures so that they can be self-contained,
at the moment, many of this figures cannot be understood at a glance.
- I do not think the choice of Figure~5 is very good: again it does not convey the main claim
of the paper: that implicit^2 is helping in terms of memory consumption. Now, if the point to
convey is that PSNR with a smaller Implicit^2 architecture can be better than with a bigger
conventional SIREN or MFN architecture, other plots or tables would be more appropriate.

Typos
----------
- l.181 correspsonding network -> corresponding network
- l.195 diemsnional -> dimensional
- l.210 this do not -> does not
- l.211 at trianing iteration -> at training iteration
- l.238 to ex scale -> ??
- l.272 achieve -> achieves

**Time Spent Reviewing:**

3

---

> ### Author Response · Authors · 2021-08-10
> **Response to Reviewer 7RDY**
>
> We sincerely thank the reviewer for the valuable feedback and questions, and for finding our work interesting. Indeed, implicit deep learning is a burgeoning branch in ML research and we are excited to see the potential benefits it can bring to INR learning. Below we provide individual responses to the reviewer’s questions and hope to clear some of the confusion.
>
> #### **1. The interpretation of Fig. 5 and the benefit of the (implicit)^2 approach**
>
> We agree with the reviewer that the current presentation of Fig. 5 is confusing and might suggest little benefit of using implicit networks for INR (a concern that is shared by reviewer YkMx). We therefore clarified in greater detail how we ought to interpret Fig. 5 in the general comment above, which hopefully could address the reviewer’s question.
>
> In short, implicit differentiation itself does not reduce memory consumption beyond the amount required by the base architecture of the iterated function (i.e., the implicit function theorem still requires backpropagation through one function evaluation). Therefore, a 1L-256D implicit model is **expected** to consume at least the same amount of memory as a 1L-256D explicit mode (which has only one layer), with the former having an implicitly defined depth and the latter having a stacked depth 1. For computational overhead, as the (implicit)^2 models re-use the fixed point, its computation cost is also expected to be equivalent to that of 1L explicit, as we highlight in Eq. (11) (notably, this is a benefit that the original DEQ model does not have). However, the comparison that we **do** want to make in Figure 5 is a deep (4L) explicit network with a 1L implicit network (e.g., 1L-512D implicit vs. 4L-256D explicit): we show that with (implicit)^2 models can substantially improve the PSNR performance by over 15%, while paying 20%-30% less cost in both computation and memory.
>
> We again refer the reviewer to the general comment 1 above to see how one should interpret Fig. 5 and how it demonstrates the significant benefit that an (implicit)^2 approach could yield. We will make sure to revise the presentation of this figure in the revision.
>
>
> #### **2. Why is (implicit)^2 approach better than SIREN or MFN?**
>
> We believe this question is closely related to the representational capacity of implicit models. In fact, the original DEQ model paper [1]  already successfully applied this kind of implicit-depth architecture to large-scale language modeling and image classification tasks, achieving results on par with the SOTA methods (e.g., Transformers); they also showed that the DEQ models are at least as expressive as explicit models (see Theorem 3 about the “universality” of [1]). Other recent works in the DL theory thread have also proved the gradient dynamics of equilibrium networks [2].
>
> Intuitively, the (implicit)^2 approach models a single layer as a dynamical system, which in itself adds complexity to the representational power of the one layer (e.g., 1L MFN). This means the (implicit)^2 approach does not predicate on any specific depth, solver, or computation graph (in contrast, a conventional network has a fixed computation graph), which is indeed an inductive bias that we hypothesize would be beneficial here (in addition to the computational and memory benefits). However, we agree with the reviewer that this is a valuable question that would be very interesting to explore as a future research direction.
>
> Finally, we agree with the reviewer that the current captions can be made more self-contained and will adjust accordingly in the final draft. We hope this response help address the reviewer's concerns and welcome any further questions.
>
> References:
>
> [1] Bai et al., Deep Equilibrium Models.
>
> [2] Kawaguchi. On the Theory of Implicit Deep Learning: Global Convergence with Implicit Layers.

---

> > ### Comment · Reviewer_7RDY · 2021-09-02
> > **Response to the authors**
> >
> > Thanks a lot for addressing my concerns. I do believe Figure~5 needs to be substantially revised as suggested by the other reviewers too and that your comment about the "representational power of dynamic system" be included in some form in the paper, with appropriate references.

---

### Official Review · Reviewer_YkMx · 2021-07-20

**Rating:** 6
**Confidence:** 4

**Summary:**

(Implicit)^2 models bring implicit computation to bear on implicit representation. Deep equilibrium network editions of the SIREN and MFN implicit representation architectures are defined and experimentally compared and contrasted with their standard explicit counterparts. The (implicit)^2 editions are optimized with two proposed approximations: reuse of fixed points during forward and truncation during backward. When these approximates are used with the proposed implicit architectures the training time and memory alike are reduced to constant w.r.t. the effective "depth" of the model as counted in fixed point iterations. This differs from explicit models, where time and memory scale with the number of layers (L), and standard use of deep equilibrium nets, where time scales with fixed point iterations (M) and memory is constant. As a further point on optimization, regularization by spectral normalization is found to be useful for (implicit)^2 training. Analysis experiments justify each approximation, in that the effects are clear in Figures 3 & 4, but the conditions of these experiments are not specified. The quality of results for explicit and implicit^2 are compared for a single input each for the three modalities of image, audio, and video. (implicit)^2 improves PSNR in all cases, and even by 10 or more percent relative for a 512x512 grayscale image and 23 percent relative for a 7-second sound. Accuracy (PSNR) and computation (memory usage in GB and training step time in sec.) are measured across three architectures for one input image: accuracy is higher or equal across model sizes, the text claims 3x faster training speed (in total?) and the text specifies memory saving though the reported charts do not (see Figure 5). The text motivates the computational attractiveness of implicit computation for implicit representations, but does not cover alternatives for reducing time or memory with explicit computation for implicit representations (like sampling/mini-batching, gradient accumulation, reversibility, model decomposition like in KiloNerf or ACORN etc.).

**Ethical Concerns:**

None.

**Limitations And Societal Impact:**

A computational limitation specific to this work is addressed by Section 5. However, the section is exclusively about this one computational limitation for storing past fixed points (state used for optimization). Discussion of potential limitations with regard to data fidelity or modality is missing, especially with respect to explicit computation for implicit representations. Regarding potential societal impacts, this work is about a generic method to improve the training of implicit representations, and as such does not have specific concerns.

**Main Review:**

Novelty:
- This is the first work to bring implicit computation to bear on implicit representation.
- Two implicit variations on implicit representation architectures are proposed, for SIREN and MFN, which are both recent and good approaches.
- The forward approximation of fixed point reuse and its experimental analysis are new, and the backward approximation of truncation is more related to other work (as discussed in the text), but the suitability of the chosen truncation (T = 1) is newly shown.

Significance:
- Implicit computation and implicit representation are both burgeoning topics that differ from the established mainstreams of explicit deep network architectures and grid/mesh/etc. representations respectively. This work joins both, and points out their potential complementarity, and so more work at this interface could follow.
- Improving accuracy (PSNR) on two strong architectures is impressive. However, the scope of the experiments is too narrow to know if the improvements are real (see next point). The quantification of the claimed computational advantages are also poorly measured/communicated: memory usage seems to be the same across explicit/implicit (Fig. 5) and differences if any in results w/ and w/o the proposed approximations are not covered.
- The results are too few to reliably measure quality w.r.t. the explicit counterparts of the proposed (implicit)^2 models. It is certainly a plus to consider three different data modalities, but that is a standard already established by prior work like SIREN, and comparisons need to be done on more than one input per modality. One example does not equal an evaluation.
- The experiments may not be properly tuned or controlled. For instance, batch size was chosen by the convenience of what would fit the largest model in memory (see supplement). Batch size and learning rate schedule can be sensitive hyperparameters, for example, so they should have been chosen by validation or used known good choices from existing works when comparing to them. Spectral normalization is applied to the (implicit)^2 models proposed, but not to the corresponding explicit models, and so results are potentially confounded by this difference.

Clarity:
- The overall organization is good, with clearly named sections, and the main conceptual figures (1 & 2) summarize the ideas.
- Key results, like the justification for the proposed forward/backward approximations, are not grounded in the concrete details of the model/data/optimization/and experimental conditions in general. None of these can be determined from the captions or text for Figures 3 and 4 (or at least I could not after several readings).
- The background on implicit computation is too vague: "designed to have no prescribed computation graph" is not quite right, as the computation graph of the iterated module is definite, but its number of iterations and the dynamics of its latents are not.
- There are a number of typos throughout, which while superficial, do nevertheless distract the reader. (See draft feedback.)

Decision:
This work's union of implicit computation and implicit representation is promising but preliminary. Trying to make the best of both is insightful, and the proposed approximations to make good on their combination are reasonable. The difference in approach is clear for implicit^2 vs. existing implicit and explicit computation, but the difference in experimental results is more potential than proven. Results are shown for a single image, a single sound, and a single video and the analysis for key properties underlying the proposed computational acceleration (fixed point reuse in Figure 3 and gradient truncation in Figure 4) are illustrative in effect but totally unexplained in experimental conditions (like architecture, data, etc.). Furthermore, it would be informative to check the use of this work's forward/backward approximations on DEQs for other purposes than implicit representations: it is fine if they work or not, but we would learn more about implicit vs. explicit. The quality of results, computational costs, and justification of the acceleration schemes are the main points of the paper and so they deserve more complete and systematic evidence.

For rebuttal:
- Please explain the actual effects on memory usage, since Figure 5 seems to indicate no memory is spared, but the text claims reductions.
- Please elaborate on the computation during training vs. testing of (implicit)^2 and explicit computation. Is it more expensive to query (implicit)^2 than explicit computations of implicit representations? Explicit could use less memory during testing in not needing the optimizer, right?
- Please report on the accuracy/computation of (implicit)^2 against the simple baseline of explicit computation with mini-batch sampling (that is, SGD on subsampled batches of input coordinates and outputs) or gradient accumulation. Does (implicit)^2 strictly dominate?

Draft Feedback
- typos: l.91 missing references with empty "[]", l. 99 "paramterized", l.111 "infintesimal", l. 195 "diemsnional", l. 203 "ponits", l. 211 "an simple", l. 211 "trianing"

**Post-Response**: The clarifications and additional results in the response have convinced me to raise my score to a 6. As the exposition and experiments still seem somewhat premature, and it seems some editing may be needed to address the nuances of the computational requirements (which are a main contribution of this submission), I cannot raise the score higher. Nevertheless the combination of implicit computation and implicit representation deserves more attention, and so this work can be considered for pointing in this direction with the first results.

**Time Spent Reviewing:**

3

---

> ### Author Response · Authors · 2021-08-10
> **Response to Reviewer YkMx**
>
> We sincerely thank the reviewer for the insightful comments. We address your concerns as follows.
>
> #### **1. Additional experiments comparing implicit and explicit models**
> Following the reviewer’s suggestions, we demonstrate the competitiveness of the proposed (implicit)^2 approach with additional experiments in 2 and 3 of the general response. The results show that a wider (implicit)^2 network is overall the best performing in these additional image representation/generalization experiments, while the implicit modeling of INR also introduces a non-trivial performance gain over explicit modeling when the base architecture is shallow. For other modalities (e.g. audio and video), we are happy to include results on additional datasets in the revised manuscript, similar to the set of experiments performed in [1].
>
> #### **2. The experimental settings for the proposed forward/backward approximations**
>
> For Fig. 3 and 4, the model and experiment configuration are equivalent to the ones used in 1L-256D Fourier-iMFN for image representation (see Sec. 4). We agree this would cause unnecessary confusion and are happy to include this discussion in the revised draft. For all other results, the detailed experiment description can be found in the supplement.
>
> #### **3. Interpretation of the computation/memory cost in Fig. 5**
>
> We acknowledge that the current presentation of Fig. 5 may cause unnecessary confusion to the reviewers. Please see 1 in our general response for a detailed discussion.  To summarize, there _are_ absolutely substantial (~2x) memory savings, and we will rework Fig 5 to emphasize this.
>
> #### **4. Training vs. testing behaviors of implicit & explicit models**
>
> During training, an explicit 1L network and an implicit 1L network have a similar level of computation overhead because they both require only evaluating the layer once (e.g., for implicit models, we re-use the fixed point), with the implicit network significantly outperforming. We do note that, as we show in Fig. 5, there is indeed a minor extra computation cost to the (implicit)^2 approach as we need to perform an additional vector-Jacobian product in the truncated backward pass. However, when we compare a still 1L (implicit)^2 model with a **4L explicit network** (which people typically use, as network depth usually leads to performance gain), we see that the computational overhead of the (implicit)^2 approach is significantly lower (along with appealing memory cost and performance).
>
> The reviewer is indeed correct that at inference time the (Implicit)^2 network cannot reuse the fixed point and still needs to go through a complete fixed-point solving procedure. Luckily, thanks to the spectral regularization we introduced, this unique fixed point can still be obtained 1) via the naive fixed-point iteration and 2) with a minimal number of steps (e.g., 6 steps empirically). Therefore, a 1L-(implicit)^2 network does not introduce a significant computational overhead over a 4L explicit network even at test time. We also highlight that as prior works have analyzed, it is possible in implicit networks to early-stop the solver to pick up inference efficiency (potentially at a cost of performance); see [2].
>
> The reviewer is also correct in that the memory cost of explicit networks can differ drastically between training and testing, as one no longer needs to store intermediate activations (e.g., of a 4L network). Whereas implicit models already enjoy a constant, one-layer memory cost benefit by formulation (and our simple fixed-point iteration does not incur additional overhead). Thus at inference time, the memory cost of explicit and implicit approaches are the same in our context.
>
> #### **5. Mini-batch (SGD) training results of the implicit models**
>
> Due to the time constraint, we only tested SGD optimization on the *Nature* dataset. To make a fair comparison, we train each of the images for 10k steps for SGD versus 2k steps for GD (As a side fact, this makes all training times of SGD longer than that of GD).
>
> | Model type \ Batch size |       64      |      256      |      1024     |      4096     |   full batch (explicit) | full batch (implicit) |
> |-------------------------|:-------------:|:-------------:|:-------------:|:-------------:|:-----------------------:|:---------------------:|
> | Gabor-MFN              | 24.07 / 23.57 | 24.85 / 24.17 | 25.35 / 24.51 | 25.66 / 24.70 | 25.78 / 24.78 | 26.66 / 25.04 |
> | Fourier-MFN            | 22.59 / 22.43 | 22.91 / 22.73 | 23.11 / 22.91 | 23.23 / 23.01 | 23.34 / 23.10 | 25.66 / 24.72 |
> | SIREN                  | 21.21 / 21.16 | 21.93 / 21.84 | 22.49 / 22.36 | 22.99 / 22.82 | 22.95 / 22.76 | 24.62 / 24.08 |
> Table 2. Mean training / testing PSNR for 1L-256D models
>
> | Model type \ Batch size |       64      |      256      |      1024     |      4096     |   full batch (explicit) | full batch (implicit) |
> |-------------------------|:-------------:|:-------------:|:-------------:|:-------------:|:-----------------------:|:---------------------:|
> | Gabor-MFN              | 24.80 / 24.06 | 25.68 / 24.65 | 26.19 / 24.95 | 26.57 / 25.12 | 26.82 / 25.21 | 29.26 / 25.53 |
> | Fourier-MFN            | 22.66 / 22.49 | 23.01 / 22.81 | 23.20 / 22.98 | 23.32 / 23.09 | 23.45 / 23.19 | 26.43 / 25.04 |
> | SIREN                  | 21.18 / 21.12 | 22.22 / 22.10 | 22.97 / 22.79 | 23.56 / 23.31 | 23.82 / 23.50 | 25.60 / 24.69 |
> Table 3. Mean training / testing PSNR for 1L-512D models
>
> | Model type \ Batch size |       64      |      256      |      1024     |      4096     |   full batch (explicit) | full batch (implicit) |
> |-------------------------|:-------------:|:-------------:|:-------------:|:-------------:|:-----------------------:|:---------------------:|
> | Gabor-MFN              | 23.44 / 23.17 | 24.69 / 24.10 | 25.80 / 24.78 | 26.66 / 25.18 | 26.20 / 24.97 | 25.99 / 24.85 |
> | Fourier-MFN            | 23.45 / 23.18 | 24.29 / 23.85 | 24.85 / 24.26 | 25.18 / 24.49 | 25.10 / 24.44 | 25.03 / 24.34 |
> | SIREN                  | 21.54 / 21.47 | 22.70 / 22.56 | 23.65 / 23.38 | 24.43 / 24.01 | 24.22 / 23.84 | 24.38 / 23.92 |
> Table 4. Training / Testing PSNR for 4L-256D models
>
> From the results, it becomes clear that a wide implicit model (1L-512D) remains the most competitive in terms of both training and testing error, while the tiniest implicit model (1L-256D) catches up in performance with the strongest explicit baselines (4L-256D w batch size 4096). Therefore, (Implicit)^2 networks still enjoy advantages over the SGD variants of explicit networks.
>
> Thank you for reading our response and please do not hesitate to engage with us in the discussion if you need additional information. We hope the reviewer could consider adjusting the score based on our response.
>
> References:
>
> [1] Sitzmann et al., Implicit Neural Representations with Periodic Activation Functions.
>
> [2] Bai et al., Deep Equilibrium Models.

---

> > ### Comment · Reviewer_YkMx · 2021-09-02
> > **Thank you for the clarifications and additional results.**
> >
> > Thank you for clarifying Figure 5 especially, and for thoroughly discussing my questions about computational cost. I appreciate the experiments on mini-batching for SGD as well.
> >
> > I do advise that you revise Figure 5, as the intended comparison is lost in its current format.
> >
> > All-in-all the response gives a clearer understanding of the experiments, and presents a more thorough discussion of the computation involved for implicit vs. explicit models of implicit representations. I have therefore raised my score to a 6. If accepted, I hope more material on the computational comparison of implicit vs. explicit computation as well as implicit computation vs. sampling in SGD can be included (whether in the main text or supplement), as this would be more informative to the community. If not, this material could be incorporated for the next round, with results on more data as provided in the response, so that the experiments are more in line with other work on implicit representations.

---

### Author Response · Authors · 2021-08-10
**General Response to All Reviewers**

We thank all the reviewers for their valuable feedback. In this general response, we hope to address some common questions shared by multiple reviewers.

#### **1. Interpretation of the computation/memory cost in Fig. 5**
We want to clarify that the primary comparisons we wish to make in Fig. 5 are between shallow **implicit** networks (i.e., the implicit 1L-256D & 1L-512D bars) and the deep **explicit** network (the explicit 4L-512D bars). For example, Fig. 5 shows that a 1L-512 (implicit)^2 Gabor-MFN model saves ~30% of the memory and ~15% of the computation when compared to a similar-sized 4L-256D explicit network, but leads to a significant ~15% performance improvement (46.8 vs. 41.4 PSNR). Similarly, even a tiny (implicit)^2 network (1L-256D) can perform on par with the explicit 4L-256 network, while being over 2x memory- and computation-efficient.

We agree that the current Fig. 5 may be confusing and will make sure to revise these visualizations and add more textual descriptions in the next draft. But in short, the remaining parts of these plots (e.g., explicit 1L-256D & 1L-512D; and implicit 4L-256D) were only meant to show that: 1) a shallow explicit network, albeit efficient, performed poorly in PSNR; and 2) implicit modeling of an already-deep explicit structure (specifically, implicit 4L-256D) does *not* significantly improve over the performance of this explicit network itself (i.e., explicit 4L-256D). Indeed, these are what we expected--- the two insights exactly reflect the benefit of (implicit)^2: by modeling a shallow single layer as a dynamical system, we are able to improve the performance of these INR networks without paying the costs that are usually induced by the explicit layer stacking.

#### **2. Additional experimental results for image representation on 100 CelebA images**

We report the PSNR on fitting the first 100 images (resized, center-cropped to 128x128) in the CelebA dataset. Each image is trained for 3000 steps with a learning rate of 3e-3.

|                          |    1L-128D   |      1L-256D     |    4L-128D   |
|-------------------------|:------------:|:----------------:|:------------:|
| Gabor-MFN               | 39.08 (3.52) |   45.85 (3.70)   | 42.55 (2.89) |
| Gabor-iMFN              | 39.39 (2.83) | **47.72 (2.02)** | 41.96 (3.17) |
| Fourier-MFN             | 35.48 (4.01) |   36.41 (4.30)   | 44.27 (2.75) |
| Fourier-iMFN            | 38.03 (2.78) | **46.70 (2.78)** | 44.72 (2.45) |
| SIREN                   | 34.41 (3.70) | 36.97 (3.98)     | 38.78 (4.06) |
| iSIREN                  | 36.75 (3.62) | **40.90 (3.08)** | 39.40 (4.07) |
Table 1. Average Testing PSNR for 100 128x128 CelebA images. Standard error is shown in parenthesis.

#### **3. Additional experimental results for image generalization on Text dataset.**

We run an additional generalization experiment on the *Text* image set, which consists of 16 512x512 images with embedded text on white background, following [1]. Other experimental conditions are identical to the image generalization experiment (4.2) in the main text.

|                          |    1L-256D   |      1L-512D     |    4L-256D   |
|-------------------------|:------------:|:----------------:|:------------:|
| Gabor-MFN               | 27.19 (2.18) |   27.74 (2.13)   | 27.57 (2.10) |
| Gabor-iMFN              | 27.53 (2.18) | **28.07 (2.00)** | 27.40 (2.10) |
| Fourier-MFN             | 24.64 (2.11) |   24.84 (2.10)   | 26.67 (2.06) |
| Fourier-iMFN            | 26.90 (2.14) | **27.19 (1.83)** | 26.48 (2.04) |
| SIREN                   | 24.54 (2.19) | 25.69 (2.18)     | 26.21 (2.19) |
| iSIREN                  | 26.06 (2.18) | **26.81 (2.09)** | 26.31 (2.20) |
Table 2. Average Testing PSNR for the *Text* dataset. Standard error is shown in parenthesis.

References:

[1] Tancik et al., Fourier Features Let Networks Learn High Frequency Functions in Low Dimensional Domains.

---

### Decision · Program_Chairs · 2021-09-27

**Decision:**

Accept (Poster)

**Comment:**

After the discussion and rebuttal process, all reviewers now recommend acceptance, albeit with several borderline ratings. Several reviewers flagged limitations of the experimental evaluation, which the authors have responded to by running additional experiments. The discussion also cleared up some confusion about memory savings, which seems to have stemmed from a misinterpretation of Figure 5.

Overall, reviewers seem to agree that the combination of implicit computation and implicit representations is interesting and creates some nice synergies, and also that this idea deserves the attention of the research community. I am therefore inclined to recommend acceptance, though this is conditional on the authors adding the new results to the paper and addressing the various other remarks and recommendations made by the reviewers (e.g. with respect to Figure 5).